# IFT88 maintains sensory function by localising signalling proteins along *Drosophila* cilia

Sascha Werner[1],* , Pilar Okenve-Ramos[1],* , Philip Hehlert[2]† , Sihem Zitouni[1,3],† , Pranjali Priya[4], Susana Mendonça[1,5], Anje Sporbert[6], Christian Spalthoff[2], Martin C Göpfert[2] , Swadhin Chandra Jana[1,4],‡ , Mónica Bettencourt-Dias[1],‡

**Ciliary defects cause several ciliopathies, some of which have late onset, suggesting cilia are actively maintained. Still, we have a poor understanding of the mechanisms underlying their maintenance. Here, we show *Drosophila melanogaster* IFT88 (*Dm*IFT88/nompB) continues to move along fully formed sensory cilia. We further identify Inactive, a TRPV channel subunit involved in *Drosophila* hearing and negative-gravitaxis behaviour, and a yet uncharacterised *Drosophila* Guanylyl Cyclase 2d (*Dm*Gucy2d/ CG34357) as *Dm*IFT88 cargoes. We also show *Dm*IFT88 binding to the cyclase´s intracellular part, which is evolutionarily conserved and mutated in several degenerative retinal diseases, is important for the ciliary localisation of *Dm*Gucy2d. Finally, acute knockdown of both *Dm*IFT88 and *Dm*Gucy2d in ciliated neurons of adult flies caused defects in the maintenance of cilium function, impairing hearing and negative-gravitaxis behaviour, but did not significantly affect ciliary ultrastructure. We conclude that the sensory ciliary function underlying hearing in the adult fly requires an active maintenance program which involves *Dm*IFT88 and at least two of its signalling transmembrane cargoes, *Dm*Gucy2d and Inactive.**

## Introduction

Cilia are microtubule (MT)-based organelles that emanate from the surface of eukaryotic cells and are vital for several functions, including motility and sensing (reviewed in Reiter & Leroux [2017]; Breslow & Holland [2019]; Sreekumar & Norris [2019]). Cilia biogenesis is a multistep process that is tightly coordinated and regulated during cell differentiation. A cilium consists of two regions: a ciliary base and a ciliary protrusion or shaft. The latter grows from the ciliary base and is composed of an MT-based skeleton (the axoneme) and a ciliary membrane. Ciliary proteins are produced in the cell body and then move either by diffusion or by active transport through the ciliary base, containing a diffusion barrier, into the ciliary shaft. A large proportion of these active transports is manifested by a process called intraflagellar transport (IFT), which depends on molecular motors moving from the ciliary base to the axoneme tip (anterograde) and in reverse (retrograde) direction (reviewed in Breslow & Holland [2019]).

Defects in cilia structure and function cause several human disorders, called ciliopathies, causing, for example, alterations in body symmetry, obesity, retinal degeneration, and cystic kidneys (reviewed in Bettencourt-Dias et al [2011]; Brown & Witman [2014]; Braun & Hildebrandt [2017]; Wang & Dynlacht [2018]; Anvarian et al [2019]). Whereas defects in cilia assembly can lead to numerous diseases, they do not account for all symptoms of many cilia-related disorders, such as retinitis pigmentosa, nephronophthisis, and Alström syndrome, which show progressive tissue degeneration or late pathological onset in the patients (reviewed in Reiter & Leroux [2017]; Sreekumar & Norris [2019]). Given that many ciliated cells such as photoreceptors, ciliated sensory neurons, and epithelial cells are long-lived, it is possible that the breakdown of ciliary maintenance is causal of those diseases. IFT has been implicated in regulating ciliary structural and functional properties such as flagella/cilia length and the ciliary localisation of signalling receptors, such as osins, transient receptor potential vanilloid (TRPV) channels, somatostatin receptors (SSTR3), and guanylyl cyclases (GC), in various organisms (Marshall et al, 2005; Qin et al, 2005; Bhowmick et al, 2009; Ye et al, 2013; Eguether et al, 2014; Jiang et al, 2015; van der Burght et al, 2020). Interestingly, in the unicellular parasite *Trypanosoma*, IFT88 depletion does not affect the structure of fully formed flagella but impairs their beating, pointing to a deregulation/mislocalisation of flagella components (e.g.,

[1]Instituto Gulbenkian de Ciência, Oeiras, Portugal    [2]Department of Cellular Neurobiology, University of Göttingen, Göttingen, Germany    [3]Institut de Génétique Humaine (IGH), UMR, 9002 CNRS, Montpellier, France    [4]National Centre for Biological Sciences- TIFR, Bangalore, India    [5]Instituto de Investigação e Inovação em Saúde, Universidade do Porto, Porto, Portugal    [6]Advanced Light Microscopy, Max Delbrück Centrum for Molecular Medicine Berlin in the Helmholtz Association, Berlin, Germany

Correspondence: swerner85@gmx.de; swadhin1cal@gmail.com; mdias@igc.gulbenkian.pt
*Sascha Werner and Pilar Okenve-Ramos shared first authors
†Philip Hehlert and Sihem Zitouni shared second authors
‡Swadhin Chandra Jana and Mónica Bettencourt-Dias shared lead authors

PKAR, kinesin 9, FAM8) that are implicated in flagellar beating (Fort et al, 2016). Furthermore, entry of several receptors, such as smoothened (SMO) and olfactory receptor coreceptor (Orco) into the cilia is IFT-independent (Milenkovic et al, 2009; Williams et al, 2014; Jana et al, 2021). Altogether, the existing evidence suggests that the maintenance of ciliary structure and sensory function may be regulated in a cell- and organism-specific manner by mechanisms that are hardly understood.

Here we study the underpinnings of ciliary maintenance in a genetically tractable organism, *Drosophila melanogaster*. In particular, we focus on adult auditory ciliated (Type-I) sensory neurons, as they rarely get replenished (Fernandez-Hernandez et al, 2021) and are involved in different sensory functions with straightforward experimental readouts, including hearing and negative-gravitaxis (Han et al, 2003; Jana et al, 2018). Given the importance of IFT in cilia assembly and maintenance in *Chlamydomonas* (Pazour et al, 2002; Marshall et al, 2005), we chose to investigate the function of the key IFT-B1 protein, IFT88, in the maintenance of cilia structure and function. *IFT88* gene is also known as *nompB* (no mechanoreceptor potential B) in *D. melanogaster* (Han et al, 2003), and we henceforth refer to it as *Dm*IFT88. We first found that *Dm*IFT88-containing trains move within fully grown chordotonal cilia (Figs 1 and 2A), suggesting IFT might play an active role in *Drosophila* cilia maintenance. We next observed that acute depletion of *Dm*IFT88 in adult ciliated sensory neurons subtly alters cilium bending at the ciliary base without affecting ultrastructures of the axoneme and impairs sensory cilia function, causing defects in hearing and negative-gravitaxis behaviour (Figs 2, S2, S3, and S4). In the search for distinct *Dm*IFT88 cargoes, we discovered that *Dm*IFT88 binds and contributes to the ciliary localisation of the TRPV channel subunit Inactive (Iav) (Fig 3) and the fly homologue of an evolutionarily conserved particulate Guanylyl Cyclase (Gucy2d) (Figs 4, 5, S5, S6, S7, S8, and S9) that is involved in human ciliopathies. Altogether, our research shows that IFT88 is dispensable for ciliary structure retention (Fig S4) but critical for the maintenance of sensory cilia function in adult *Drosophila*, partly through the binding and localisation of different signalling proteins, including Inactive and *Dm*Gucy2d.

# Results

## IFT88 protein sequences show high conservation of structural domains

*D. melanogaster* homologues of IFT proteins (i.e., A-complex, B-complex, and BBSome) have been identified through bioinformatics analysis (for a summary, see our compilation of all information in Fig S1A, and Table S1). Several of these proteins were shown to be expressed in ciliated neurons or in the fly head (Avidor-Reiss et al, 2004; Chintapalli et al, 2007; Lee et al, 2008, 2018; Mourao et al, 2016), yet only a small number have been investigated mechanistically, such as *Dm*IFT88 and *Dm*IFT140 (Han et al, 2003; Lee et al, 2008). Studies in other model organisms suggest that only a few evolutionarily conserved IFT proteins are critical for cilia formation (Pazour et al, 2002; Fan et al, 2010; Eguether et al, 2014; Fort et al, 2016). Among them is IFT88, and thus, we investigated whether it is active

after the cilia assembly is complete and might also have an important role in transporting components needed for cilium maintenance.

In *D. melanogaster*, two isoforms of *Dm*IFT88 have been described with similar amino acid compositions (Fig 1Ai) (Han et al, 2003). Both isoforms are predicted to bear 10 tetratricopeptide repeat (TPR) domains (Karpenahalli et al, 2007) (Fig 1Aii), which often act as interfaces for protein-protein interactions (Allan & Ratajczak, 2011; Taschner et al, 2012), and, accordingly, seem good candidates for transporting cargoes into the cilium. In fact, all IFT88 homologues (in insects and vertebrates) are predicted to comprise 10–15 TPRs (Fig 1Aii; for a detailed list of protein accession numbers, see Tables S2 and S3), pointing to conserved roles in mediating protein interactions, either within the IFT-B core complex or between IFT and various cargoes (Taschner et al, 2012). We thus investigated whether *Dm*IFT88 transports cargoes and continues to do so after cilium assembly, contributing to cilia maintenance.

## *Dm*IFT88 trains are visible in fully assembled cilia

To assess the localisation of *Dm*IFT88 protein in fully assembled cilia, we first tested a transgenic fly line expressing GFP-tagged *Dm*IFT88 protein (GFP::*Dm*IFT88) under its endogenous promoter (Han et al, 2003). Yet, the weak signal resulting from GFP-tagged *Dm*IFT88's expression did not allow for live imaging. We thus generated transgenic lines carrying a UAS-eGFP::*Dm*IFT88 (isoform-RD) construct, as this isoform suffices to rescue IFT88 function in *nompB*[1] (a *Dm*IFT88 functional null mutant) flies in which all isoforms are affected (Han et al, 2003). To test whether *Dm*IFT88 is part of IFT trains in assembled cilia, the transgene was expressed using a chordotonal neuron-specific driver (Gal4[*Iav*]) (Fig 1Bi–iii) (Gong et al, 2004). We were able to track *Dm*IFT88 particles in the proximal compartment of the cilia of lateral chordotonal organ (lch5) neurons in wandering L3 larvae, a developmental stage at which larval ciliogenesis is considered to be concluded. Quantification of the velocity of the eGFP::*Dm*IFT88 signal (Fig 1Ci and Video 1) revealed that the particles move about five times faster in the retrograde direction (1.2 $\mu$m/s) than in the anterograde direction (0.22 $\mu$m/s). A recent study on the chordotonal neurons in developing *Drosophila* pupae found different anterograde (~0.44 $\mu$m/s) and retrograde velocities varied (~0.12 and ~0.7 $\mu$m/s in proximal and distal compartments, respectively) for IFT88 (Lee et al, 2018). These observations suggest that *Dm*IFT88 velocities vary between developmental stages and ciliary compartments. This large variability was also shown before for different cell types in single species (Besschetnova et al, 2010; Williams et al, 2014; an overview of different IFT velocities is provided in Table S4).

We also noticed that the signal intensities of the anterograde trains of eGFP::*Dm*IFT88 appear stronger than the retrograde train intensities, similar to features observed in other organisms, including *Chlamydomonas* and *Trypanosoma* (Fort et al, 2016; Wingfield et al, 2021). Measurements of *Dm*IFT88 train lengths (Fig 1Cii), although limited by the resolution of the confocal microscope, suggest that anterograde trains have an average length of 358 nm (ranging from 152–650 nm as minimum and maximum length). This would correspond to trains with ~58 IFT-B particles, based on a recent report in *Chlamydomonas* (van den Hoek et al, 2022). We found that retrograde

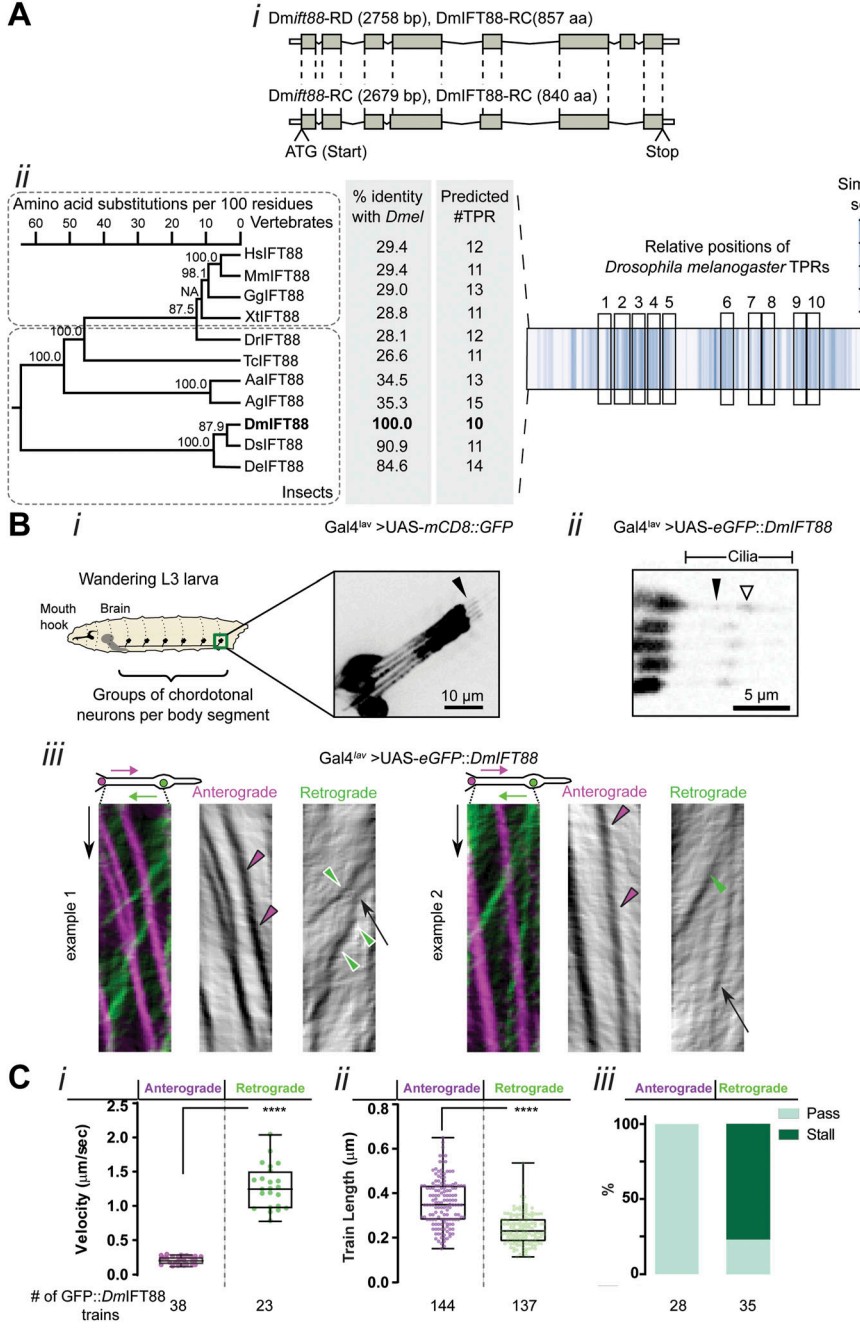

**Figure 1. *Dm*IFT88 is evolutionarily conserved and its trains are visible in *Drosophila* sensory cilia.**
**(A)** *Drosophila* IFT88 shows structural-domain conservation despite low amino acid sequence conservation. (i) Schematic representation of the two annotated *DmIFT88* isoforms (RNA and proteins) in the fly. The grey boxes represent coding sequences. (ii) Left: Maximum-likelihood phylogenetic tree for IFT88 from various vertebrate and insect species, displaying bootstrap branch "support values" in percentages (%). The accession numbers of the proteins used in this analysis and a list of abbreviations are provided in Table S2. NA: the "support value" could not be calculated because of the method used to generate the sequence alignment. The amino acid identity of each sequence compared to *Drosophila melanogaster* is shown in percentages (%), and the number of predicted tetratricopeptide repeat domains in each species is displayed. Right: Multiple sequence alignment of IFT88 from 11 species represented as a heat map generated using JProfileGrid2. Each position in the alignment is shown as a box, colour-coded according to the similarity score. The relative positions of the 10 tetratricopeptide repeats of *Drosophila melanogaster* are indicated by black boxes. **(B)** GFP::*Dm*IFT88 trains are visible in wandering L3 larvae. (i) Schematic representation of a L3 larva showing the segmentally arranged groups of chordotonal neurons (a group of five neurons is called lch5). Membrane-bound GFP (UAS-mCD8::GFP) is expressed using Gal4$^{Iav}$ to visualise the morphology, including cilia (arrowhead), of one such group (lch5) of neurons. (ii) A video still showing ectopically expressed GFP::*Dm*IFT88 in the dendrite tip and cilia in lch5 neurons using Gal4$^{Iav}$. Empty and filled arrowheads mark ciliary dilation and IFT particles along the proximal cilium, respectively. (iii) A scheme of a cilium showing the IFT particles moving in anterograde (magenta) and retrograde (green) directions. Below are two examples of merged kymographs with both types of *Dm*IFT88 particle trains colour-coded depending on their direction. On the right of each merged kymograph example, separated grey anterograde (first) and retrograde (second) kymographs are shown. The train-tracks were extracted using the *Kymograph Clear* macro toolset from ImageJ. Magenta arrowheads, green arrowheads and arrows indicate the anterograde, retrograde and stalled trains, respectively. **(C)** (i) Left: Quantifications of the speed of the *Dm*IFT88 particles, extracted from videos from 5 larvae from three different experiments. (ii) Quantifications of the lengths of *Dm*IFT88 particles in the proximal part of lch5 cilia. (iii) Percentages of visualised trains that pass or pause when encountering an opposite train along the proximal region of the cilium. In (C), *P*-values are calculated using Mann-Whitney test (****P*-value ≤ 0.0001) on Prism. Source data are available for this figure.

trains are significantly shorter, with an average of 237 nm (ranging from 115–433 nm as minimum and maximum length), corresponding to ~38 IFT-B particles (Fig 1Cii). Although IFT trains in the fly have not been studied in detail by electron microscopy (EM), these numbers are very close to those found in other species by EM or cryo-ET, supporting previous indications that, compared to retrograde trains, anterograde trains contain more IFT complexes and move in a tighter conformation (Pigino et al, 2009; Buisson et al, 2013; Fort et al, 2016; Chien et al, 2017; Jordan et al, 2018).

In *Chlamydomonas* flagella, a microtubule doublet is considered a double track for IFTs, with retrograde trains moving along A-microtubules and anterograde trains moving on B-microtubules (Stepanek & Pigino, 2016). Whereas IFT movement has not been as well detailed in *Drosophila,* our analysis of *Dm*IFT88 train movements revealed interesting features. We observed that anterograde trains move slowly and remain unperturbed by any encounter with retrograde trains. In contrast, retrograde train movements are irregular (Fig 1Biii), half of them stalling their trip at least once along the length of the proximal

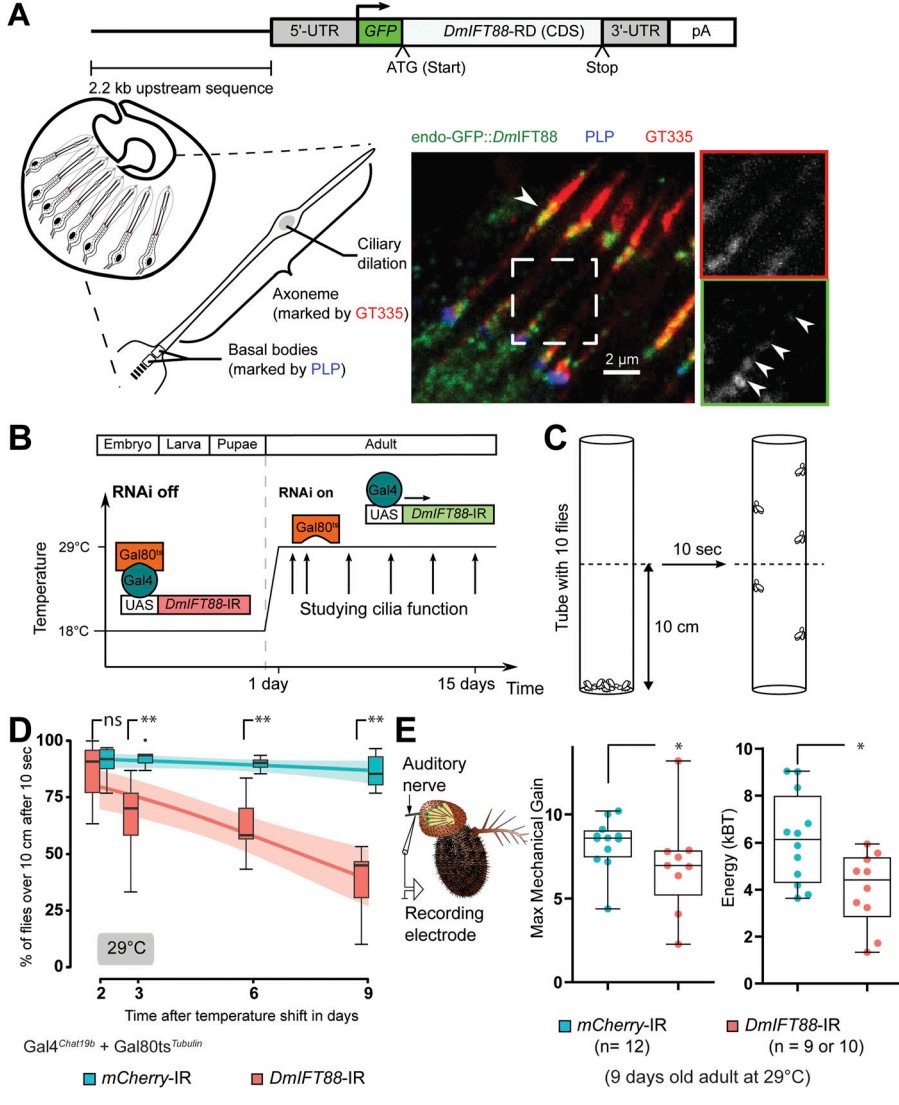

**Figure 2. Acute removal of *DmIFT88* in adult flies leads to impaired sensory functions.**
**(A)** *Dm*IFT88 protein is found in fully formed cilia in the adult. Top: Representation of the transgene (from [Han et al [2003]](ref)), expressing GFP::*Dm*IFT88 near endogenous level, used to observe the *Dm*IFT88 localisation in fly cilia. Left: A scheme of the chordotonal neuron architecture in the second antennal segment of adult flies showing the expected localisation of *Drosophila* Pericentrin-like protein and glutamylated tubulin (GT335) in the basal bodies and the axoneme, respectively. Right: representative image of the localisation of endogenous GFP::*Dm*IFT88 with respect to the two aforementioned markers in the adult chordotonal cilia. Note that GFP::*Dm*IFT88 signals were enhanced with an anti-GFP antibody. Arrowheads mark GFP::*Dm*IFT88 puncta at the ciliary dilation and along the axoneme (inset). **(B)** Scheme of the approach and timeline of the conditional knockdown (*DmIFT88* and *mCherry* RNAis) experiments. *DmIFT88* and *mCherry* genes are knocked-down in cholinergic neurons, including chordotonal neurons, using Gal4^Chat19b. Flies are reared at 18°C to repress the expression of the hairpin during development through the co-expression of a temperature-sensitive version of Gal80 ubiquitously (TubGal80^ts). After flies come of pupae, adult flies are shifted to 29°C (non-permissive temperature for Gal80^ts) to activate the expression of RNA hairpins. **(C)** Schematic representation of the climbing (negative-gravitaxis) behavioural assay. The effect on sensory cilia function is approximated by quantifying climbing behaviour (number of flies reaching a certain height in a specific time) on a controlled setup of the adult flies on specific days (arrows) after temperature shift. **(D)** Time-dependent changes in climbing behaviour at 29°C in control (*mCherry*) and *DmIFT88* RNAi flies under Gal4^Chat19b and Gal4^Iav drivers (left and right, respectively). Each box-plot corresponds to a total of 60 flies measured in sets of 10 animals each. The data are fitted using linear regression, where the area around the curve represents the 95% confidence interval. The two lines are significantly different at 29°C. **(E)** Scheme of the set-up of the electrophysiology experiments performed in the 9 d old flies' antennae after analysis of the hearing nerve function (left).

Electrophysiology data (right): All-range box plots of antennal fluctuation powers (Energy) and maximum sensitivity gain to mechanical amplification (Maximum Mechanical Gain) of the hearing nerve responses (median ± min and maximum) the flies with genotypes stated in the graphs legend. In (D, E), *P*-values are calculated using Mann-Whitney test (ns - *P*-value › 0.05, *P*-value ≤ 0.05 and **P*-value ≤ 0.01) in Prism.
Source data are available for this figure.

compartment of the cilium, often when encountering an anterograde train. Indeed, when two particles moving in opposite directions encounter each other, all the anterograde particles analysed continued their trip without delay and no change in velocity, whereas most of the retrograde particles (76%) temporarily paused before resuming their movement towards the base of the cilia (Fig 1Ciii).

To explain these results, we propose that either in these cilia opposite direction trains can compete for the same protofilaments/microtubules, or in the case they always run on separate MTs, there is some cross talk between the tracks that regulate retrograde IFT. Interestingly, dyneins (–ve end) are more flexible motor proteins with the ability to switch protofilaments and have a larger step size than kinesins (+ve end) (Ross et al, 2008). It is thus possible that in case of an encounter of opposite-directed trains, dynein-mediated retrograde trains (and not kinesin-powered

anterograde) are the ones that switch protofilament/MT tracks within a doublet. A further detailed analysis of the IFT features in *Drosophila* would be required to uncover underlying mechanisms of these intriguing IFT features.

Given that *Dm*IFT88 moves along fully formed chordotonal cilia of larvae, we wondered whether it might also be present in chordotonal cilia of adult flies as found in Johnston's organ (JO), the hearing organ in the fly's second antennal segment (Fig S1B). We detected both *Dm*IFT88 isoforms (Fig S2B) in mRNA isolated from adult antennae (Fig S2C), suggesting this gene continues to express after ciliogenesis. Because live imaging was not possible in the adult, we immuno-stained antennal sections of flies expressing GFP::*Dm*IFT88 under the endogenous promoter and observed an enrichment of the protein at the ciliary base and at the ciliary dilation of the chordotonal neurons (Fig 2A). The strong signals near

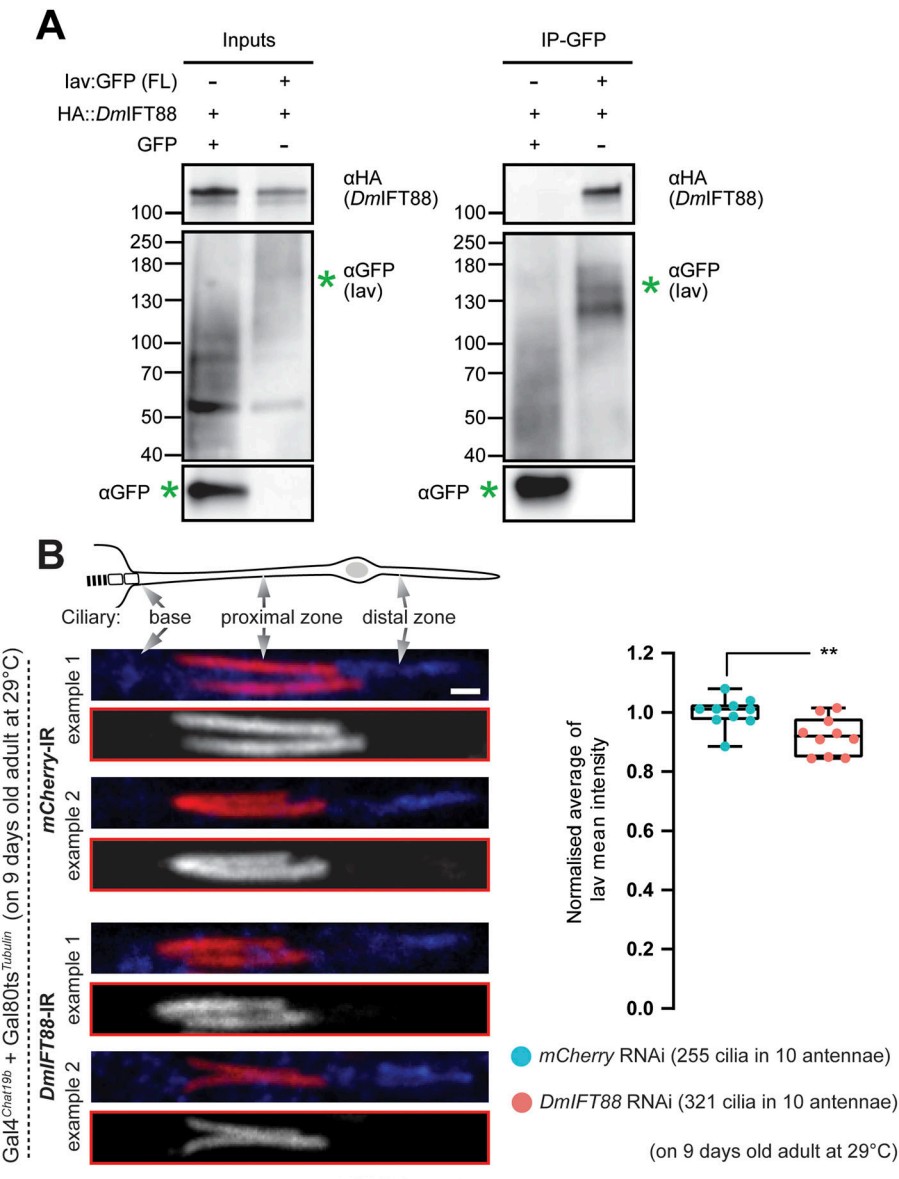

**Figure 3. Inactive, a ciliary TRPV channel, binds *Dm*IFT88 and requires it for its localisation.**
**(A)** Co-immunoprecipitation (co-IP) assay performed upon co-overexpression of 3xHA::*Dm*IFT88, and Inactive::GFP (or GFP as negative control) in *Drosophila* Dmel cells. *Dm*IFT88 co-immunoprecipitates with Inactive::GFP but not with GFP alone (N = 3 repeats). The green asterisk marks the expected band of the transfected Inactive::GFP plasmid. The rest of bands seen with the GFP antibody were found in all the three independent experiments and most likely are because of possible phosphorylation or to unwanted degradation (for details on the antibodies used, see Table S7).
**(B)** Left: Representative immunofluorescence images of adult chordotonal cilia with Inactive (Iav) and NOMPC, in the proximal and distal zone of the cilia, in red and blue, respectively. Upper two examples are from control flies (*mCherry* RNAi); below two examples are from *DmIFT88* RNAi flies. For details on the antibodies used, see Table S7. Scale bar: 1 μm. Right: All-range box plots of the normalised average Iav signal (per antennae) along the proximal part of the axoneme in flies with the aforementioned genotypes. *P*-value is calculated using Mann–Whitney test (**P*-value ≤ 0.01).
Source data are available for this figure.

the ciliary base and the dilation may represent an immobile fraction of the *Dm*IFT88 pool (Fig 1Bii and Video 1). Additional smaller and weaker GFP puncta, similar to the ones observed in the larvae (Fig 1Bii, filled arrowhead), were detected along the axoneme, likely representing IFT trains (inset on Fig 2A). Together, our observations on larvae and adult flies (Figs 1Biii and 2A) suggest that IFT88 features observed in the fly are to some extent evolutionarily conserved and that *Dm*IFT88 is actively transported along the chordotonal cilia beyond ciliogenesis.

### Conditional knockdown of *DmIFT88* in adult cilia impairs hearing and gravitaxis behaviour

We next investigated whether *Dm*IFT88 has any significant role in the maintenance of cilia. Because this protein is essential for

ciliogenesis in sensory neurons (Han et al, 2003), we could not take advantage of the available mutant, as it does not assemble cilia. We thus aimed at knocking down *DmIFT88* expression in a certain subset of sensory neurons using RNAi (Table S5), which was only active after the cilia had formed.

We first tested for the efficacy of the RNAi line (with a hairpin that targets an exon common to both *DmIFT88* isoforms and without any predicted off-targets) using a ubiquitously active promoter (Gal4$^{Tubulin}$). We found that the mRNA levels of both isoforms are strongly reduced (~75–80%) in the antennae of flies expressing the *DmIFT88* hairpin (UAS-*DmIFT88*-IR), as compared to negative controls (UAS-*mCherry*-IR, as *mCherry* is not encoded in the fly genome; Fig S2A–C). Importantly, sound stimulation-evoked compound action potentials recorded from JO chordotonal neurons were significantly reduced in *DmIFT88* knockdown flies compared to

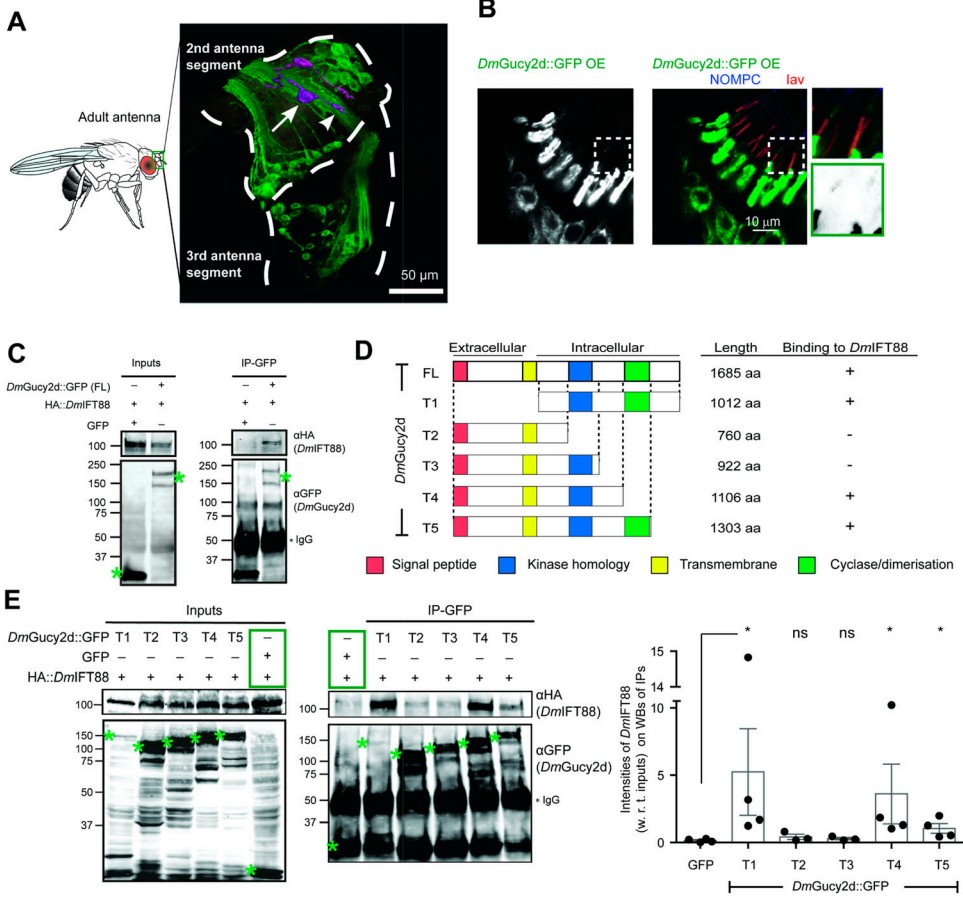

**Figure 4. CG34357, a *Drosophila* homologue of Gucy2d, localises to chordotonal cilia and it binds *Dm*IFT88 through its intracellular portion.**
**(A)** Using the enhancer line to drive UAS-mCD8::GFP expression, *CG34357* expression is detected in the antennae of adult flies. The image stack was converted into a 3D model using Imaris software (see Video 2). The outline of the second and third antenna segment is drawn using the autofluorescence of the cuticle. Two chordotonal neurons in the second antenna segment are highlighted in magenta indicating the cell body (arrow) and the dendrite (arrowhead).
**(B)** Immunofluorescence image shows that although the ectopically expressed *Dm*Gucy2d::GFP protein (using Gal4*Iav*) primarily accumulates in the dendrites of the adult chordotonal neurons, it also localises to the cilia. Insets (2x of the dashed boxes) show that the GFP signal can also be detected in the ciliary dilation (distal to the Iav protein signal). **(C)** Co-immunoprecipitation (co-IP) assay performed upon co-overexpression of 3xHA::*Dm*IFT88, and *Dm*Gucy2d::GFP (or GFP as control) in *Drosophila melanogaster* (Dmel) cultured cells. *Dm*IFT88 co-IP with *Dm*Gucy2d::GFP but not with GFP alone. **(D)** Left: Schematic representation of *Dm*Gucy2d truncation constructs used to determine its binding region to *Dm*IFT88, showing also the domain structures and constructs length (aa). Right: A summary of the various *Dm*Gucy2d truncated peptides' ability to bind *Dm*IFT88 (from (E)) is shown. **(E)** IP assay performed upon co-overexpression of full-length 3xHA::

*Dm*IFT88 and GFP-tagged fragments of *Dm*Gucy2d in Dmel cells. Right: Bar plots (overlaid with scattered dots) of the fraction of *Dm*IFT88 (with respect to the inputs) binding to the GFP-tagged fragments of *Dm*Gucy2d and GFP alone. Each bar contains fractions of bound *Dm*IFT88 intensity values (= values in the IP/values in the Input) measured on the western blots, for at least three independent experiments. In (C, E), co-overexpressed GFP and 3xHA::*Dm*IFT88 serves as a negative control in the experiments (green boxes). Note that we also detect a faint non-specific HA-positive band in the IPs from extracts that co-express GFP and 3xHA::*Dm*IFT88 and anti-GFP antibody detects a few non-specific bands in all input and IP lanes. The bands of expected molecular weight are marked with green asterisks. For details on the antibodies used, see Table S7. In (E), *P*-values are calculated using Mann-Whitney tests (ns - *P*-value > 0.05, *P*-value < 0.05).
Source data are available for this figure.

controls (Fig S2D), consistent with the hearing defects reported in *DmIFT88/nompB* null mutants (Han et al, 2003). Interestingly, we found that whereas flies with the ubiquitous *DmIFT88* knockdown do not assemble cilia, they can still build the transition zone (Fig S2E–H), similar to the phenotype observed in IFT88 and kinesin-2 mutants in various model organisms (Pazour et al, 2002; Sarpal et al, 2003).

To test for defects in cilium maintenance, we used a temperature-sensitive (ts) system that allows us to control gene expression in a cell- and time-specific manner (Gal4-UAS-Gal80^ts) (Fig 2B; see the "Material and Methods" section) (McGuire et al, 2003). We used the Gal4^Chat19b driver, which expresses in the peripheral and central nervous system, including JO neurons during development (Salvaterra & Kitamoto, 2001). In antennal tissue, this driver expresses at the pupal stages, long after neurogenesis begins, and a few hours after ciliogenesis begins. It continues to be expressed in adult flies ([Lienhard & Stocker, 1991; Jana et al, 2011]; Figs S3A and S4A). The TubGal80^ts system (at 18°C, TubGal80^ts is active, thus there is no RNAi knockdown; at 29°C, TubGal80^ts is inactive, thus

knockdown through RNAi is active) allowed us to acutely down-regulate *DmIFT88* expression with a RNAi line just during adult maintenance by moving to and rearing flies at 29°C after flies emerge from the pupae (Fig S3A). This experimental setup further grants us to completely disentangle *DmIFT88* described function in ciliogenesis (Han et al, 2003) from its possible role in adult cilia maintenance.

Because gravity sensing in *Drosophila* requires functional chordotonal neurons, we analysed the fly's negative-gravitaxis behaviour by a modified version of a well-characterised climbing assay (used earlier in Jana et al [2021]) at different ages, for up to 9 d after induced RNAi expression (reducing ~30% *DmIFT88* mRNA levels compared to control flies; Fig S3C) because the climbing behaviour of the flies older than 9–10 d (at 29°C) could be affected by natural ageing in the adults (Figs 2B and C, S3E, and S7D; [Kamikouchi et al, 2009]). Importantly, *DmIFT88* RNAi flies developed negative gravitaxis defects in 3 d old flies (reared at 29°C) that, with time, became more pronounced (Fig 2D). When flies were reared at 18°C (Gal4 inhibited), there was no significant negative-

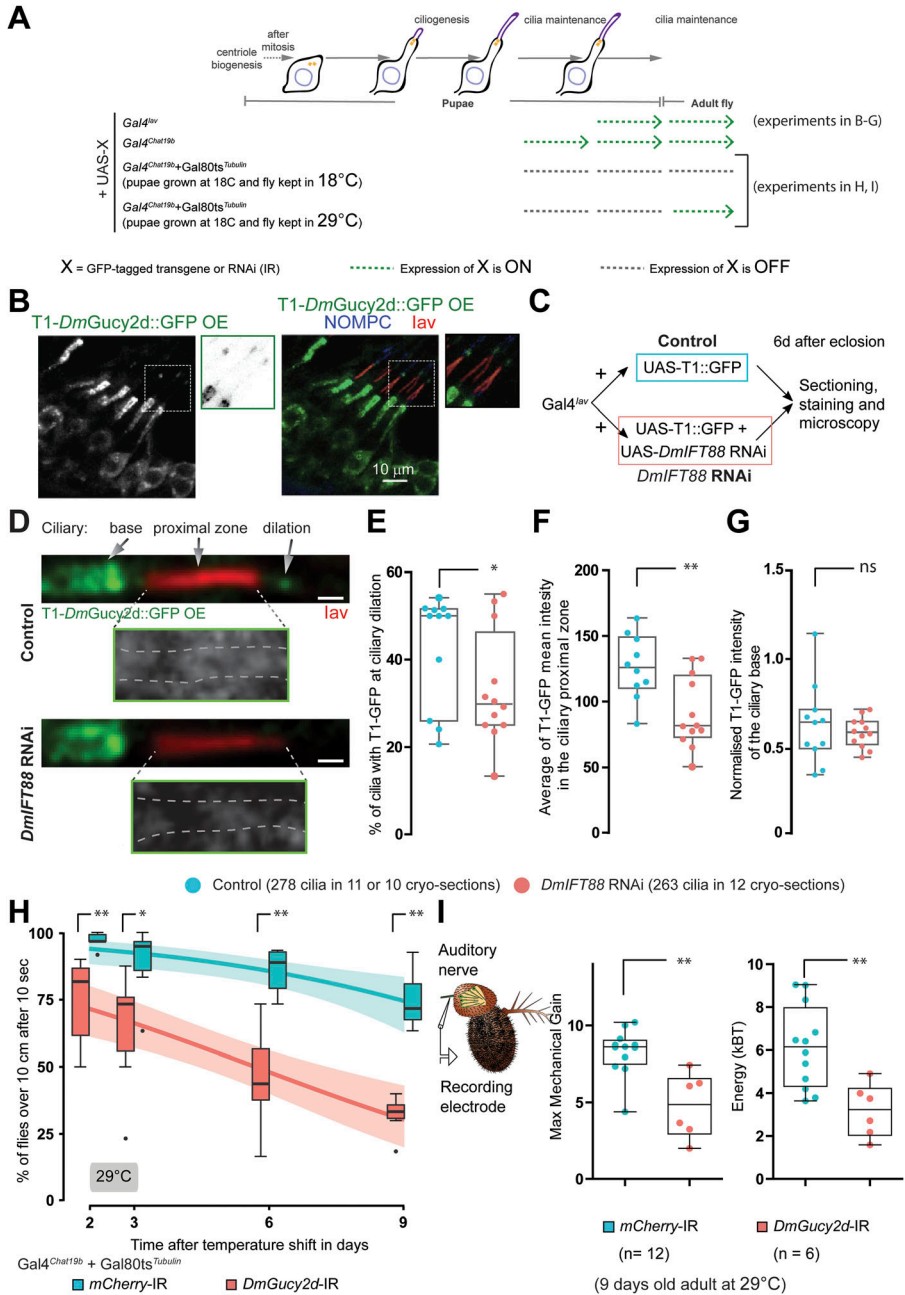

**Figure 5. _Dm_Gucy2d is localised in the cilium via _Dm_IFT88, and it is important for the maintenance of behavioural sensory functions.**
**(A)** Representation of the drivers' expression in the experimental settings used in the following sections of the figure as mentioned in the scheme. **(B)** Immunofluorescence images show that the ectopically expressed GFP-tagged T1-truncation of _Dm_Gucy2d (using Gal4^_Iav_) accumulates in chordotonal neuron cell body, dendrite and ciliary dilation. Insets highlight that the GFP signal is also seen in the cilium axoneme. Proximal and distal cilia zones are marked with Iav (red) and NOMPC (blue) antibodies, respectively. **(C)** Scheme summarises the experimental strategy in which T1-_Dm_Gucy2d::GFP is expressed in the chordotonal neurons (using Gal4^_Iav_) with or without RNAi against _Dm_IFT88. The resulting adult flies were analysed 6 d after they came of pupae. **(B, D)** Representative immunofluorescence images of the adult chordotonal cilia from flies with different genotypes (described in (B)). Insets show T1-GFP localisation along the proximal zone of the cilium (marked with dashed grey lines based on the anti-Iav antibody signal). Scale bars: 1 μm. **(D, E, F, G)** Graphs are all-range box plots of: the percentage of cilia with GFP signal at the ciliary dilation (D), the average GFP signal along the proximal zone of the cilium (E), and normalised signal intensities of GFP at the ciliary base (F). Here, _P_-values are calculated using Welch corrected (One-tailed) unpaired _t_ test (ns - _P_-value ˃ 0.05, *_P_-value < 0.05, **_P_-value ≤ 0.01). **(H)** Negative gravitaxis behaviours of flies during time at 29°C carrying the corresponding control (_mCherry_ RNAi) and _DmGucy2d_ RNAi, under the expression of the Gal4^_Chat19b_-TubGal80^ts driver. Each box plot corresponds to a total of 60 flies measured in sets of 10 animals each. The data are fitted using linear regression on the left panel, where the area around the curve represents the 95% confidence interval. The two lines are significantly different at 29°C. **(I)** Scheme of the set-up of the electrophysiology experiments performed on the 9 d old flies' antennae to analyse hearing nerve function (left). Electrophysiology data (right): All-range box plots of antennal fluctuation powers and maximum sensitivity gain because of mechanical amplification (maximum mechanical gain) of the hearing nerve responses (median ± min and maximum) the flies with genotypes stated in the graphs legend. In (H, I), _P_-value is calculated using Mann-Whitney test (*_P_-value < 0.05, **_P_-value ≤ 0.01). Source data are available for this figure.

gravitaxis behaviour impairment at any time point tested, compared to control flies (Fig S3B). To assess whether this behavioural phenotype is because of the loss of _DmIFT88_ in chordotonal organs only in adults, the conditional knockdown experiment was repeated using a temperature-sensitive system with a more restricted promotor for Gal4 expression, _Gal4^Iav_. This driver is expressed only in chordotonal neurons, but it is also weaker, which forced us to extend the study up to 14 d. The analysis of this condition also showed that upon knockdown of _DmIFT88_, there is a significant impairment on climbing behaviour (Fig S3E).

In addition to impairing negative-gravitaxis, acute _DmIFT88_ knockdown in cholinergic neurons (Fig 2E) negatively affected the

function of the cholinergic chordotonal neurons of the fly's JO on hearing. These neurons transduce antennal vibrations into electrical signals and, in addition, actively amplify these vibrations on a cycle-by-cycle basis through ciliary motility (Gopfert & Robert, 2003; Nadrowski et al, 2008; Karak et al, 2015). This motility was strongly reduced by _DmIFT88_ knockdown (after 9 d), as witnessed by a drop of the mechanical sensitivity gain and a reduced power of the antenna's mechanical free fluctuations (Fig 2E). Altogether, these results document the importance of _Dm_IFT88 for maintaining sensory function after ciliogenesis is completed.

To investigate whether sensory function loss in the acute _DmIFT88_ knockdown flies is because of defects in the ciliary/

axoneme structure, we studied the ultrastructure of chordotonal neuron cilia in JO. Strikingly, cilia were always present, but in 9- and 15-d post-knockdown flies, the curvature of the base of cilia mildly increased compared to controls (Fig S3D). Although in larval chordotonal neurons, ciliary dysfunction has been associated with a bending of the cilia (Zanini et al, 2018), the significance of this morphological phenotype in sensory function in adults is still unclear.

To further check to what extent IFT88 is required for the maintenance of the ciliary structure, we knocked down *DmIFT88* under the Gal4$^{Chat19b}$ driver, but without the use of the TubGal80$^{ts}$ system (Fig S4A). In this condition, we expected to further increase the depletion of the *DmIFT88* gene in a ciliary maintenance context. Indeed, we found similar levels of mRNA knockdown to those previously observed with the constitutive driver Gal4$^{Tubulin}$, which avoided entire ciliary axoneme formation (Fig S2C) (66 ± 20% RNA knockdown for Gal4$^{Chat19b}$ versus 71 ± 12% RNA knockdown for Gal4$^{Tubulin}$) (Fig S4C). Despite Gal4$^{Chat19b}$ *DmIFT88 RNAi* flies displaying a strong gravitaxis behaviour phenotype (Fig S4B), the number of cilia per scolopale, gross ciliary features (such as transition zone and axoneme), and ultrastructural properties (e.g., the number of doublets and inner and outer dynein arms) remained largely unaffected in these Gal4$^{Chat19b}$ *DmIFT88* knockdown flies (even after 15 d of knockdown) when compared to controls (Fig S4D–F). These results further confirm *Dm*IFT88's important role in cilia function maintenance without a key implication on cilia ultrastructure maintenance.

## Inactive, a TRPV calcium channel subunit, is a *Dm*IFT88 cargo

Loss of chordotonal cilia function in conditional knockdown of *DmIFT88* suggests that *DmIFT88*-dependent cargo transport is continuing even after ciliogenesis is complete. Given the acute knockdown of *DmIFT88* did not affect the axonemal ultrastructure, including the number of MT doublets and dynein arms (Fig S4D–F), we hypothesised that the deterioration of the ciliary function of chordotonal neurons might reflect an impaired transport of signalling proteins along the cilia. Only a few signalling proteins have been characterised in the chordotonal cilia of the fly. One interesting signalling protein is the TRPV channel subunit Inactive (Iav). TRPVs belong to the conserved TRP superfamily of cation-selective channels. They are cation transporter channels, highly calcium selective, and have multiple functions as sensory mediators in various tissue types and species (reviewed in Baylie & Brayden [2011]; Liu et al [2022]). In particular, Iav (*Drosophila* orthologous to human TRPV1-6) resides in the proximal cilium region of fully formed chordotonal cilia and is required for locomotion and hearing (Gong et al, 2004; Sun et al, 2009; Zhang et al, 2013; Kwon et al, 2020). Moreover, IFT is implicated in the localisations of several TRP channels along the ciliary shaft in *Chlamydomonas* and *Caenorhabditis elegans* (Qin et al, 2005; Huang et al, 2007). In particular, loss of other IFT components (e.g., rempA/IFT140) or retrograde IFT motors (btv/cytoplasmic dynein heavy chain) during cilia assembly in *Drosophila* leads to defects in Iav localisation (Lee et al, 2008; Kwon et al, 2020), proposing that Iav could be an IFT-dependent ciliary cargo.

Therefore, to test this hypothesis, we first explored whether *Dm*IFT88 interacts with Iav. We used *D. melanogaster* (Dmel) cultured cells, which normally express neither of these proteins nor most of the IFT proteins (Hu et al, 2017), including *Dm*IFT88, making them an appropriate system to test for protein-protein interactions. We performed co-IP experiments using Dmel cells that ectopically and transiently expressed both *pUAS-3xHA::DmIFT88* (this study) and *p-UAS-Iav::GFP* (kindly provided by Yun Doo Chung [Gong et al, 2004]) or *p-UAS-GFP* under an ubiquitous Gal4$^{Act5C}$ driver. 3xHA::DmIFT88 specifically bound to Iav::GFP but not to cytoplasmic GFP control (Fig 3A), strongly suggesting Iav is an interactor of *Dm*IFT88.

To further test whether Iav is a cargo of *Dm*IFT88, we acutely knocked down *DmIFT88* in chordotonal neurons of the antennal JO and assessed Iav localisation using an anti-Iav antibody (Gong et al, 2004). In the *DmIFT88* knockdown flies, although Iav continued to localise in the proximal compartment of the chordotonal cilium, the channel's concentration was mildly but significantly reduced (by ~9%, median) compared to the control (Fig 3B). Altogether, these data strongly suggest *Drosophila* Iav is a cargo of *Dm*IFT88 at least in adult hearing ciliated neurons.

The small reduction of Iav along the proximal region of the cilium upon *DmIFT88* RNAi knockdown seems unlikely to account for the strong negative-gravitaxis and hearing defects seen in the *DmIFT88* RNAi flies. Moreover, *Iav* knockout increases ciliary motility and, thus, the mechanical amplification provided by JO neurons (Gopfert et al, 2006), whereas in *DmIFT88* RNAi knockdown flies, this amplification was reduced. Therefore, we hypothesise that the sensory deficits caused by *DmIFT88* knockdown, rather than being caused by the mislocalisation of one cargo alone, may reflect the combined phenotype of the mislocalisation of several *Dm*IFT88 indirect and direct cargoes, with Iav being one of them.

## CG34357, an orthologue of mouse Gucy2e, localises to chordotonal cilia and binds to *Dm*IFT88 by its intracellular domain

In mouse photoreceptors, a hypomorphic mutation in *IFT88* exhibited retinal degeneration (Pazour et al, 2002), and in humans, heterozygous mutations in *IFT88* give rise to inherited retinal degeneration (Chekuri et al, 2018). This suggests that mutations in some of its cargoes might cause similar phenotypes. Indeed, several TRP channels are expressed in retinal cells, and their dysfunction is also implicated in retinal degenerative pathologies (reviewed in Thebault [2021]). In addition, vertebrate Gucy2d/e, a membrane-bound (particulate) cGMP-generating enzyme, alternatively known as Guanylyl Cyclase (GC), is required for photoreceptor function. *Gucy2e* mutations are in fact implicated in human retina-specific ciliopathies that can be related to the development/maintenance of ciliary function, such as Leber congenital amaurosis (LCA) and retinitis pigmentosa (Zagel & Koch, 2014). Importantly, in IMCD3 cells, IFT88 was shown to transport ectopically expressed mouse Gucy2e to primary cilia (Bhowmick et al, 2009). The synthesised cGMP is a second messenger of signal transduction essential for several sensory functions such as olfaction and vision. Components of this primordial signalling pathway, which include GCs, phosphodiesterases (PDE), and protein kinase G (PKG), are known to regulate a wide range of sensory functions, such as nociception,

olfaction, and phototransduction, in various animals (reviewed in Johnson & Leroux [2010]; Wen et al [2014]; Maruyama [2016]). Not surprisingly, cGMP signalling components are found in vertebrate and invertebrate organisms located within cilia (Morton, 2004; Johnson & Leroux, 2010), although this signalling has not been yet associated with the function of *Drosophila* cilia. Therefore, we investigated the evolutionary conservation of the mouse Gucy2e (Table S3) and searched for its *D. melanogaster* homologue.

The *Drosophila* genome encodes several particulate membrane GCs (Morton, 2004), but none of them has so far been associated with gravitaxis behaviour or sensory cilium function. Using the protein sequence of the mouse Gucy2e (an orthologue of human Gucy2d) in PSIBlastp search (Altschul et al, 1997) against the *D. melanogaster* protein database, we identified several putative particulate GCs (Fig S5A). Because cGMP signalling components are important for sensory behaviour in several organisms and cell types (Johnson & Leroux, 2010), we reasoned that knockdown of a GC in chordotonal organs might impair cGMP generation in neurons, thus the negative-gravitaxis behaviour. Fly stocks bearing RNAi constructs for six putative GCs (four particulate and two soluble) were available, and we used them to knockdown the respective genes using Gal4$^{Chat19b}$ (for expression pattern of this driver, see Fig S4A) and to perform climbing assays for each of them. Our results showed that the knockdown of three of the four particulate transmembrane candidates caused behavioural defects (Fig S5A), whereas the predicted soluble GCs did not. The closest *Drosophila* homologue of mouse *Gucy2e*, the protein product of which was reported to be transported by IFT88, is the uncharacterised gene *CG34357*, and knockdown of this gene leads to negative-gravitaxis behaviour defect in adult flies (Fig S5A, B, and D). Thus, for follow-up studies, we focused on it. According to the human gene nomenclature, we renamed *CG34357* as *DmGucy2d*.

To further pursue our analysis of *DmGucy2d*, we first investigated the expression profile of this gene. The information publicly available suggests that *DmGucy2d* gene is expressed in the PNS and in CNS (Graveley et al, 2011; Brown et al, 2014), and we tested its endogenous expression in chordotonal neurons. For that purpose, we used an enhancer Gal4-trap line (NP0270) to express a membrane-bound GFP reporter (UAS-mCD8::GFP) (Fig S6A and B). We observed that the enhancer is expressed in the chordotonal neurons in both L3 larvae (Fig S6C) and the adult second antennal segment (Fig 4A). We detected GFP signal only in a subset of adult chordotonal neurons, suggesting that the enhancer trap line might not fully recapitulate the endogenous gene expression or that only a subset of chordotonal neurons expresses *DmGucy2d*. Notably, we also observed expression of *DmGucy2d* in Type-III dendritic neurons. Therefore, for further studies, we expressed or perturbed this gene only using specific drivers expressed only in a subset of sensory neurons (e.g., Gal4$^{Chat19b}$ and Gal4$^{Iav}$ express in cholinergic and chordotonal ciliated neurons, respectively), without compromising *DmGucy2d* expression and protein localisation in other cell-types. To further assess *DmGucy2d*'s subcellular protein localisation, we cloned the coding sequence of its longer isoform (RD/RC) and generated transgenic flies. The protein was tagged with GFP at the C-terminus to avoid cleavage of the tag because of a predicted N-terminal signal peptide (Figs 4D and S5B and C). When UAS-*DmGucy2d::GFP* was expressed using a chordotonal neuron-specific driver (Gal4$^{Iav}$), GFP fluorescence was observed in the dendrites and along their cilia (Figs 4B and S6D).

To test whether *Dm*Gucy2d might be a *Dm*IFT88 cargo, we examined if *Dm*Gucy2d binds to *Dm*IFT88 using *Drosophila* cultured cells (Dmel), as previously done to study interactions between *Dm*IFT88 and Iav (see Fig 3A). Note that both *DmGucy2d* or *DmIFT88* do not express in these Dmel cells (Hu et al, 2017). We co-expressed either *Dm*Gucy2d::GFP (this study) or cytoplasmic GFP alone, together with HA-tagged *Dm*IFT88. Unlike the GFP control, *Dm*Gucy2d::GFP (full length, FL) binds to HA::*Dm*IFT88 (Fig 4C).

A typical particulate GC has several protein domains (Figs 4D and S5B and C): (i) a signal peptide at the extracellular domain; (ii) a short transmembrane domain; (iii) a kinase homology domain important for the interaction with other intracellular proteins, such as the GC-E activator (the membrane-anchored activating protein, activated by Ca$^{2+}$); (iv) a homodimerisation domain important for the activation of the catalytic domain; and (v) a cyclase catalytic domain, which synthesises the cGMP that can in turn activate ion channels or be used by PKG or PDE. The predicted protein product of *DmGucy2d* has all these features (Fig S9B). To narrow down the *Dm*IFT88 binding region of *Dm*Gucy2d, we generated truncated constructs of the cyclase (Fig 4D). The T1 fragment (T1-*Dm*Gucy2d::GFP, Fig 4D) exclusively comprises the entire intracellular part of the protein. The other four fragments (T2–T5) all contain the extracellular and transmembrane domains that might be necessary for membrane localisation and various truncations of the intracellular domain. Each of these (T2–T5) fragments is successively longer towards the intracellular C-terminus (Fig 4D). Our co-immunoprecipitation experiments revealed that T1, T4, and T5 co-immunoprecipitate *Dm*IFT88 (Fig 4E; three or more experimental repeats per construct were plotted in the graph and statistically analysed), suggesting that the interaction requires the intracellular portion encountered mainly between the kinase domain and the cyclase/dimerisation domains. Because neither Iav nor *Dm*Gucy2d nor most of the IFT proteins, including *Dm*IFT88, are expressed in Dmel cells (Hu et al, 2017), our results suggest that *Dm*Gucy2d-*Dm*IFT88 and Iav-*Dm*IFT88 interactions are direct. Still, we cannot entirely rule out an intermediate player in assembling these complexes.

Our data also strongly suggest that the intracellular domain of *Dm*Gucy2d plays a key role in *Dm*Gucy2d localisation, presumably through its transport by *Dm*IFT88. To test whether this is true, we investigated: (i) whether GFP-tagged full-length and T1 (entire intracellular domain) fragment of the cyclase move along the cilia in similar manner as *Dm*IFT88; and (ii) the effect of knockdown of *DmIFT88* on the localisation of GFP-tagged T1-*Dm*Gucy2d along the cilia. We extensively attempted to live-image *Dm*Gucy2d::GFP and T1-*Dm*Gucy2d::GFP in the lch5 neurons of L3 larvae. Both *Dm*Gucy2d::GFP and T1-*Dm*Gucy2d::GFP, when overexpressed (using Gal4$^{Iav}$) in a WT background, were highly enriched in the dendrite and the base of the cilia with a membranous and particle-like appearance, albeit show fainter localisation along the proximal segment and at ciliary dilation of the chordotonal cilia (Fig S6D). These results suggest this cyclase is transported up the cilia dilation and accumulated there. However, despite of extensive trials, we failed to observe

particulate movement along the ciliary shaft for either GFP-tagged *Dm*Gucy2d or T1-*Dm*Gucy2d (each live-imaging was of at least for 3 min, n = 28 Ich5 organs from 8 larvae analysed on five experimental days; n = 20 Ich5, from 6 larvae analysed on four experimental days, respectively), suggesting unlike *Dm*IFT88 the *Dm*Gucy2d does not move into and along the cilia continuously, at least during our imaging period (Figs 5B and S6D empty arrowheads). Furthermore, these results prevented us from investigating the role of *Dm*IFT88 in live transport of *Dm*Gucy2d along the cilia and limited our experiments to fixed adult tissues (Figs 4B and 5B).

In *DmIFT88* knockdown adult flies (Fig 5A and C), the percentage of cilia with T1-*Dm*Gucy2d::GFP signal at the ciliary dilation was strongly reduced (by ~40%, median) compared to controls (Fig 5D and E). In addition, the mean intensity of T1-*Dm*Gucy2d::GFP was significantly reduced (by ~38%, median) at the proximal region of the cilium (Fig 5D and F). At the cilium base, however, GFP fluorescence remained unaltered (Fig 5D and G), suggesting that *Dm*IFT88 specifically regulates T1-*Dm*Gucy2d localisation in the ciliary shaft but not its localisation at the dendrite tip/ciliary pocket.

We next tested whether *Dm*Gucy2d is important for maintaining cilium function in chordotonal neurons. We used the conditional inducible promoter system Gal4$^{Chat19b}$-TubGal80$^{ts}$ to express the hairpin RNA against *DmGucy2d* in adult ciliated chordotonal neurons (see Fig S8A and B for RNAi knocking-down efficiency). *DmGucy2d* RNAi knockdown slightly reduced negative-gravitaxis at 18°C (Fig S7A and B), possibly reflecting leaky expression of the construct. Importantly, after the temperature shift to 29°C, compared to control flies, those with *DmGucy2d* knockdown developed mild, albeit significant climbing defects (Fig 5H). Similar mild climb defects were found with the use of the more restricted driver, Gal4$^{Iav}$ (Fig S7A and C), and with the use of another RNAi (KK110863, VDRC) line under the same acute Gal4$^{Chat19b}$ conditions (Fig S7D and Table S5). This finding further documents that *Dm*Gucy2d in adult chordotonal neurons is needed for the maintenance of the proper sensory input and thus maintains the normal negative gravitaxis behaviour. In addition to affecting gravity sensing, 9-d acute RNAi of *DmGucy2d* resembled the auditory defects caused by knockdown of *DmIFT88*, showing reduced maximum sensitivity gain and energy of the antenna's mechanical free fluctuations (Fig 5I). Hence, like *Dm*IFT88, *Dm*Gucy2d is required for maintaining normal hearing in adult flies.

Finally, we further characterised the relationship between *Dm*IFT88 and the two cargoes identified in this work. With that purpose, we first studied the ciliary bending in acute *DmGucy2d* knockdown flies. Unlike *DmIFT88*, *DmGucy2d* conditional knockdown did not alter the angle of the chordotonal cilia at their bases (Fig S8C), suggesting this *DmIFT88* knockdown phenotype is not linked to *Dm*Gucy2d. Moreover, the acute knockdown of *DmGucy2d* in adults did not change Iav mean intensity along the proximal region of the cilia in the chordotonal organ at 9 and 15 d after the adult flies emerge (Fig S8D), suggesting ciliary localisation of Iav is independent of *Dm*Gucy2d. These results further support the idea that *Dm*IFT88 transports several cargoes, whose localisation and functions at the cilium may or may not be dependent on one another, and that are together involved in the maintenance of ciliary function.

# Discussion

Cilia biogenesis depends on the activity of an exclusive transport system called IFT (Pazour et al, 2002; Eguether et al, 2014; Jiang et al, 2015), yet whether IFT also has a role in maintaining metazoan cilia has been little explored (Marshall et al, 2005; Hao et al, 2011; Fort et al, 2016). Here we show that *Dm*IFT88, an evolutionarily conserved core IFT-B1 complex component (Fig S1), is present in chordotonal neurons and is continuously mobile along the length of mature sensory cilia (Fig 1). Acute depletion of *DmIFT88* after ciliogenesis in ciliated neurons leads to impaired sensory function and behavioural defects without severely affecting axoneme ultrastructure, suggesting an involvement of IFT in sensory cilium function beyond its known pivotal role in ciliary axoneme assembly (Figs 2, S2, S3, and S4). We hypothesised that impairment of the localisation of cargoes, directly or indirectly transported by *Dm*IFT88, would explain the observed function defects upon the IFT88's conditional knockdown. A few ciliary transmembrane signalling molecules were considered IFT-cargoes in various eukaryotes (reviewed in Lechtreck [2015]), but no *Dm*IFT88-dependent direct cargoes in the fruit fly were known. Here, we identify two *Drosophila* IFT88-dependent cargoes: Iav, a TRPV Ca$^{2+}$ channel subunit involved in sound and gravity sensing (Fig 3); and the transmembrane protein *Dm*Gucy2d (*CG34357*), which we characterise in this study and whose homologues serve multiple signalling roles in eukaryotes. We further demonstrate that *Dm*IFT88 binds to the intracellular domain of *Dm*Gucy2d (Fig 4). Mutations in this region of the cyclase are implicated in human diseases (Figs S5 and S9). Finally, we show that depletion of this cyclase in mature sensory neurons impairs negative-gravitaxis behaviour and hearing, substantiating its role in *Drosophila* auditory ciliary function maintenance (Figs 5, S6, S7, and S8).

## *Dm*IFT88 plays differential roles in cilia assembly and maintenance

Multiple mechanisms have been implicated in the regulation of cilia homeostasis in diverse model organisms (Marshall et al, 2005; Hao et al, 2011; Jiang et al, 2015; Fort et al, 2016). Ciliary structure and composition have to be maintained, and they are likely to be dynamic and altered in response to external stimuli, as was shown for *C. elegans* sensory cilia (DiTirro et al, 2019). Studies in *Chlamydomonas*, worms, and mice suggest that continuous tubulin turnover at the ciliary tip is required to maintain flagellum/cilium length (Marshall et al, 2005; Hao et al, 2011; Jiang et al, 2015). In contrast, in fruit flies, even though *Dm*IFT88 is pivotal for axoneme assembly ([Han et al, 2003], Fig S2E–H) as in other organisms (Pazour et al, 2002; Kohl et al, 2003), we show that it has no major role in maintaining axoneme ultrastructure and only a minor influence on maintaining ciliary gross morphological features (Figs S3 and S4). We found here that *DmIFT88* knockdown in adults, after ciliogenesis is complete, leads to a decrease in sound-evoked action potentials and impaired negative-gravitaxis, which suggests that sensory cilia function (i.e., auditory and gravity-sensing-related transductions) during adulthood may require a maintenance program. These results show that *Dm*IFT88, a core evolutionarily conserved

component of the IFT-B1 complex, plays distinct roles at different developmental times within even the same cell type. Note that our result is similar to what was reported in *Trypanosoma*, suggesting that although IFT88's role in ciliary skeleton homeostasis is not conserved among different ciliated species and tissues, this protein plays an evolutionarily conserved role in the maintenance of ciliary sensory function (Fort et al, 2016). Further studies, in the future, are required to investigate whether other evolutionarily conserved IFT proteins are involved in ciliary structure homeostasis.

### Both *Dm*IFT88 cargoes, Iav and *Dm*Gucy2d, are involved in sound and gravity sensing in the adult flies

Chordotonal ciliated neurons are important to sense external mechanical stimuli (i.e., wind, sound, and gravity) during adulthood essential for the fly to escape, mating and foraging (Hehlert et al, 2021). Chordotonal JO's neurons are exquisitely sensitive, amplifying sound-induced mechanical vibrations, even if they hardly exceed Brownian motion, to facilitate their detection (Nadrowski et al, 2008). Although several transmembrane and signalling proteins have been implicated previously to be IFT-associated cargoes in other organisms (Qin et al, 2005; Huang et al, 2007; Williams et al, 2014), neither homologues of those molecules were shown to be essential for fly sensory cilia functions nor the signalling molecules implicated in *Drosophila* JO neurons functions were associated to IFT. Our data suggest that the transport of various ciliary transmembrane signalling novel cargoes by *Dm*IFT88, including Iav and *Dm*Gucy2d, in fully formed sensory cilia is critical for the homeostasis of the ciliated chordotonal neurons' functions (Figs 3–5).

Mechanotransduction in *Drosophila* chordotonal organs involves TRP channels (Hehlert et al, 2021), probably because of the role of the cations exchanged to transduce external stimuli to the cell (Li et al, 2021). In particular, several subfamilies of $Ca^{2+}$ TRPV transporters are found in the cilia of sensory neurons, being essential for several sensory functions, including chemosensation, proprioception, and hearing in a variety of species (Gong et al, 2004; Bargmann, 2006). In *Drosophila*, the TRPV subunit Iav, an essential pore subunit for auditory transduction, is expressed in chordotonal organs and localises to the proximal region of the cilium (Gong et al, 2004; Li et al, 2021). Mutants for *Iav* are deaf (Gong et al, 2004) and show defective negative-gravitaxis (Sun et al, 2009). The low (9%) Iav protein intensity decrease found upon *Dm*IFT88 knockdown during adulthood (~30% *Dm*IFT88 RNA decrease) (Figs 3 and S3) might be because of either a slow Iav turnover along the auditory cilia or a *Dm*IFT88-independent process that is also involved in maitaining some fraction of axonemal Iav at the adult stage. These hypotheses remain to be further investigated.

*Dm*Gucy2d, a particulate GC, could contribute to the signal amplification of auditory transduction in multiple ways. On the one hand, the cGMP that it synthesises might be used by the ciliary machinery (i.e., cGMP is directly used by ion channels, PKGs, or PDEs) after being activated by low-level external stimulation, as it happens in mouse photoreceptor cells where $Ca^{2+}$ regulates the activity of Gucy2d (Johnson & Leroux, 2010; Wen et al, 2014; Maruyama, 2016). On the other hand, it might also be involved in modifying the mechanical

properties of the cilium in response to stimulation, by adjusting the activities of dynein arms present at the proximal region of the auditory cilium that are critical for ciliary motility and sound sensitivity (Karak et al, 2015; Li et al, 2021).

Recently, IFT88 was described to interact with the leucine-rich repeat containing 56 protein (LRRC56) when both were ectopically co-expressed in HEK293 mammalian cells (Bonnefoy et al, 2018). Similarly, we found both Iav and *Dm*Gucy2d independently bind to *Dm*IFT88 in Dmel cells, an in vitro system where none of these three proteins express in a physiological condition. To fully appreciate the potential of IFT in regulating various ciliary properties, such as structural maintenance, plasticity, composition, and function, it will be important in the future to identify additional cargoes transported by IFT88 and other IFT proteins. Moreover, to fully understand how cilia are maintained, it will be interesting to uncover whether and how IFT-independent transport could play a role in that process.

### Evolutionarily conservation of Gucy2d structure and function and its possible implications in human retinal diseases

Several mutations spread along the entire sequence of the guanylyl cyclase Gucy2d are implicated in retinal disorders, including LCA (Fig S9C). LCA is a family of congenital retinal dystrophies that results in severe vision loss due, in some cases, to the inability of the phototransduction cascade to take place in the patients' eyes (reviewed in den Hollander et al [2008]; Tsin et al [2018]). Little is known about the mechanisms by which those mutations cause the disease. Mutations affecting either the cyclase activity (Jacobson et al, 2013), protein stability, or transport into the cilia can potentially be harmful. Although several results indicate that *Gucy2d* gene therapy replacement could be a good treatment for this family of diseases, to date, there is no cure or treatment for these patients (reviewed in Chacon-Camacho & Zenteno [2015]).

Our research uncovered that the intracellular domain of the cyclase, which we found to be essential for its localisation in the *Drosophila* sensory cilium, is evolutionarily conserved (Fig S9A and B). Furthermore, we discovered that several of the human Gucy2d residues found to be modified in some LCA patients in the intracellular region are conserved in *Drosophila* Gucy2d (Fig S9C and Table S6) (Tucker et al, 2004; Li et al, 2011; Jacobson et al, 2013; de Castro-Miro et al, 2014; Zagel & Koch, 2014; Feng et al, 2020; Liu et al, 2020; Salehi Chaleshtori et al, 2020). Our work suggests a novel role of the evolutionarily conserved intracellular domain of *Dm*Gucy2d, which is to bind to *Dm*IFT88 for the cyclase to be appropriately localised/transported along the cilia. We hypothesise that the possible defective transport of Gucy2d along human photoreceptor cilia is one of the causes of LCA disease not contemplated before.

## Materials and Methods

### Materials

Find lists of accession numbers for IFT88 proteins from different organisms, accession numbers of GC proteins from various

species, antibodies (primary and secondary) and their dilutions, primers, and the genotypes of the flies used in this study in Tables S2, S3, S5, S7, and S8, respectively.

## Methods

### *Bioinformatics analysis*

For IFT88, the gene model for *DmIFT88* was extracted from the Ensembl Genome Browser (Yates et al, 2016). The number of TPR domains was predicted using the TPRpred tool (Karpenahalli et al, 2007). For phylogenetic analysis, IFT88 protein sequences of 11 metazoan species were used (see Table S2). Whenever several *IFT88* isoforms were reported for one species, the largest one was chosen for analysis. The MegAlign program (Lasergene suite, Version 8.1.3, DNASTAR) was used to generate a phylogenetic tree summarising sequence similarities as a function of the number of amino acid substitutions between sequences. Sequences were aligned using the ClustalW algorithm with default settings. Bootstrapping analysis was performed with default values to calculate the "support value" of each branching point in the tree (Fig 1Aii left). Multiple sequence alignments are presented as a heatmap. An alignment file in the FASTA format using the web browser-based MUSCLE tool was created (Edgar, 2004) to extract sequence identities. Subsequently, the ProfilGrid tool (Roca et al, 2008) was used with default settings to assign a similarity score to each position in the alignment, with the score assuming values between 0 and 1. The results are presented as a heatmap, which was visualised with RStudio (RStudio) and the ggplot2 package (Wickham, 2016) (Fig 1Aii right).

For guanylyl cyclases, including *DmGucy2d/CG34357*: Sequence analysis for guanylyl cyclases (Fig S5A) was performed as described above. For the accession number of guanylyl cyclase protein sequences used in this analysis, see Table S3. The gene model for *CG34357* was taken from the Ensembl Genome Browser (Yates et al, 2016).

### Cryo-sectioning, immunolabeling, and image analysis

Histological sections of the adult antenna were prepared as described before (Mishra, 2015; Jana et al, 2016). If not stated otherwise, preparation steps were carried out at room temperature. Flies were anaesthetised using $CO_2$ and decapitated. The heads were collected in pre-cooled fixation buffer (4% PFA, 75 mM PIPES buffer [pH 7.6], 0.05% Triton-X, picric acid) and incubated for 40 min. Good fixation requires the heads to sink to the bottom of the reaction tube. The heads were washed three times in PBST (PBS with 0.05% Triton-X) and then incubated on a rotator in PBST with 10% sucrose, first for 1 h at room temperature and then, overnight, in 25% at 4°C. The heads were embedded and oriented appropriately in OCT (optimal cutting temperature formulation; Tissue-Tek; Sakura Finetek), and the OCT was frozen on dry ice. Head samples were sectioned at 12 $\mu$m thickness in a Leica Cryostat CM 3050 S (Leica). The slices were collected on poly-L-lysine-coated slides (Sigma-Aldrich).

Sections were washed with PBST (PBS with 0.05% Triton-X) three times for 10 min each, followed by incubation in blocking buffer (10% BSA in PBST) for 1 h and subsequent incubation with the

primary antibody diluted in blocking buffer overnight at 4°C. Samples were washed three times in blocking buffer and incubated for 1 h with secondary antibody in blocking buffer. Finally, after three additional washing steps using PBS for 5 min each, samples were embedded in Vectashield supplemented with DAPI (Vector Laboratories). 12-bit images were acquired on a TCS SP5 upright confocal microscope (Leica) using an PL APO CS 63.0 × 1.40 OIL objective with a pinhole at 1 Airy unit and a pixel size of 80–200 nm (lateral) and 400 nm (axial). For details of the antibodies used, see Table S7.

To quantify in the *DmIFT88* knockdown experiments the signal intensities of Inactive (Iav; a marker for the proximal cilium) antibody staining and the GFP signal of the T1-*Dm*Gucy2d::GFP at the proximal cilium (Fig 3B, 9-d old flies; and Fig 5D, 6-d old flies), image stacks were analysed with the Imaris software v.9.2.1 (Bitplane). For analysis and generation of Fig 3B: the Iav channel was used to generate segmented volume surfaces, and the Iav intensity values were retrieved from Imaris, averaged per antennae in Excel (Microsoft Office), and normalised to the negative control of each repeated experiment. For graphs in Fig 5E–G: the Iav channel was used to generate segmented volume surfaces. Next, mean intensity values of Iav antibody staining and T1-*Dm*Gucy2d::GFP at the segmented Iav surfaces were retrieved from Imaris, averaged per antennae in Excel (Microsoft Office), and statistically analysed (non-parametric Mann-Witnney *t* test or Welch Two Sample *t* test) in Prism (GraphPad). Data were represented on box plots (median ± full range, each dot representing individual averaged values) using Prism 5.2.

To quantify the T1-*Dm*Gucy2d::GFP intensity at the end of the dendrites in the adult chordotonal neurons (Fig 5E), the image stacks were deconvolved using the Huygens Deconvolution v17.4 software (Scientific Volume Imaging) with an CMLE algorithm followed by image analysis in Imaris (as mentioned above). This deconvolution step facilitated the segmentation of the crowded dendrite population in the adult antennae by increasing the signal-to-noise ratio (SNR) (N > 10 antenna in all conditions).

### mRNA isolation from adult antenna and reverse transcription analysis

Between 100 and 200 flies were decapitated per genotype. Heads were collected in Protein LoBind tubes (Eppendorf) on dry ice. 10–20 heads were collected at a time to avoid compromising the tissue because of long exposure to room temperature. Tubes were snap-frozen in liquid nitrogen. Antennae were dissociated from heads through mechanical force using a vortex mixer (three times for 10 s, cooled in between again on dry ice). To separate antennae from heads, tubes were opened and inverted. Heads will fall out, whereas most of the antennae will adhere to the walls of the tubes. Total mRNA was isolated from antennae using the PureLink RNA Mini Kit (Ambion Life Technologies). Antennae were lysed in 250 $\mu$l of lysis buffer. The protocol was carried out as described in the accompanying manual of the kit. No DNAse treatment was performed. The mRNA was always converted to cDNA immediately. For cloning, 1 $\mu$g of total mRNA was reverse transcribed. The reverse transcription reaction was performed with the Transcriptor First Strand cDNA Synthesis Kit (Roche). For quantifying mRNA levels,

50–70 heads were used. For cloning *DmIFT88*, cDNA was generated using poly-dT primers included in the kit. The cDNA samples were stored at –20°C until further use.

## Cloning and generation of transgenic flies

For *DmIFT88*, the cDNA for cloning *DmIFT88* was obtained from mRNA isolated from antennae of $w^{1118}$ flies. *DmIFT88* (isoform-RD) was amplified using the KOD polymerase kit (EMD Millipore) in 50-µl reaction volume (for PCR primers, see Table S8). PCR was set up as outlined for the kit. The reaction mix was supplemented with DMSO. The PCR product was purified using the DNA Clean & Concentrator kit (Zymo Research) and adenylated at the 5′ ends through incubation with DreamTaq (Thermo Fisher Scientific) using the appropriate buffer and dNTPs at 70°C for 1 h. Subsequently, the PCR product was ligated into pGEM-T Easy (Promega) following the manufacturer's instructions, and plasmids were confirmed through sequencing (for sequencing primers, see Table S8). Using the Gateway system, the coding sequence was first cloned into pDONR 221 in a BP-reaction (Invitrogen), following the instructions for the kit, and subsequently transferred to pTHW and pTGW in an LR-reaction (Invitrogen) to fuse the resulting protein to three HA-tags or one GFP-tag on the N-terminus, respectively (for details on pTHW and pTGW, see Drosophila Genomic Resource Centre). *Dm*IFT88 was tagged at the N-terminus because the GFP:: *Dm*IFT88 fusion protein was previously shown to rescue the *DmIFT88/nompB* mutant phenotype and can hence be considered fully functional (Han et al, 2003).

For *DmGucy2d/CG34357* full length (FL) and truncated T1-*Dm*Gucy2d GFP constructs, the *CG34357* coding sequence was cloned from mRNA isolated from the antennae of $w^{1118}$ flies, as described above. *DmGucy2d* could only be amplified from cDNA produced with random hexamer primers (included in the Transcriptor First Strand cDNA Synthesis Kit; Roche). For primer sequences, see Table S8. To enhance the chance to amplify the desired target (coding sequence of *CG34357*-RD, Fig S4B) even further, the mRNA was reverse-transcribed using a primer on the 3′-untranslated region of *DmGucy2d* (*CG34357*_3UTR_rev 2 µM in a 20 µl reaction volume, 1 µg of total mRNA were used as template). Two rounds of PCRs were performed to obtain the *DmGucy2d* with the appropriate overhangs for Gateway cloning (Invitrogen). In the first round, the *DmGucy2d* coding sequence was amplified using primers located on each of the untranslated regions. For the second PCR, 2 µl of unpurified PCR product of the first PCR was used as a template. Primers containing appropriate sequences for gateway cloning were used for the second PCR. The coding sequence was cloned without the stop codon to allow for gene fusion at the C-terminus. Tagging at the N-terminus was not considered because the protein is predicted to undergo cleavage of the signal peptide (30 amino acids, Fig S5B, black arrowhead). The coding sequence was confirmed through sequencing and sub-cloned into pTWG (Drosophila Genomic Resource Centre) for C-terminal fusion with GFP.

Plasmid DNA was amplified in *E. coli* DH5α and purified using ZR Plasmid MiniprepTM-Classic (Zymo research). ZymoPURE Midiprep Kit (Zymo research) was used for large-scale DNA purification that is required (*DmIFT88*, *DmGucy2d*, or T1-*DmGucy2d* in pTWG) for transfection experiments and fly transgenesis. Injections into *Drosophila* embryos and selection of positive transformants were outsourced (BestGene or IGC transgenics facility) (see Table S5 for detailed genotypes of flies used).

Detection of *DmGucy2d* gene expression: to detect expression of *DmGucy2d* in different developmental stages, an enhancer Gal4-trap line (*CG34357*[NP0270]) was obtained (Brand & Perrimon, 1993). Flies carrying the enhancer trap insertion were crossed with flies encoding 40xUAS-IVS-mCD8::GFP (see Table S5 for detailed genotypes of flies). The mCD8::GFP fusion protein is targeted to the membrane of the cells and was used to visualise the morphology of cells. 40 UAS tandem insertions were chosen to increase the signal intensity (see Fig 4B).

For imaging adult antenna, the cuticle of the heads was cleared, as further explained below, and the second antenna segment was imaged entirely on an SP5 Live Upright microscope (Leica). Confocal Z-stack images were deconvolved using the Huygens v17.4 software (SVI) and 3D reconstructed using the Imaris v6.4 software (Bitplane), which was also used to prepare the video (Video 2). Wandering L3 larvae were mounted and imaged as described for the live-imaging of L3 larvae below.

## Live imaging of chordotonal cilia in L3 larvae and measurement of IFT88 train lengths

GFP:: *Dm*IFT88 was expressed in chordotonal neurons using Gal4$^{Iav}$ (Gong et al, 2004; see Table S5 for detailed genotypes of flies). Flies were reared at 25°C to the third larval stage, and all experiments were performed with wandering L3 larvae. Larvae were collected and briefly washed in PBS. For imaging, larvae were immobilised between a coverslip and a slide in a drop of PBS (Zhang et al, 2013). The cover slip was held in place using transparent double-sticky tape. Imaging was performed using a Roper spinning disc microscope (Nikon). Samples were kept at 25°C through the imaging process. To prevent artefacts from cell death, larvae were always imaged immediately after immobilisation and for a maximum duration of 2 min. Images were acquired every 120 ms with 100 ms exposure time (Figs 1B and S6). Images were analysed using the FIJI software (Schindelin et al, 2012), and kymographs were generated using the Kymograph-Clear plugin (Mangeol et al, 2016).

To obtain the IFT train lengths, we used these kymographs (examples shown in Fig 1Biii) showing separately *Dm*IFT88 anterograde and retrograde trains. To measure the approximate lengths of the trains, we used the Straight Line and Measure Tools from Fiji to draw and measure the thickness, and thus the length, of the trains (a total of 30 trains in both directions were measured). Data plot and statistical analysis were done using Prism (GraphPad).

## Quantification of *DmIFT88* expression levels from adult antennae

For measuring *DmIFT88* expression levels, the total mRNA was isolated from the antennae of adult flies with specific genotypes

up to 4 d after adult flies emerged from pupae using real-time PCR (qRT–PCR) (see Table S5 for detailed genotypes of flies). The web browser-based Primer-BLAST tool (Ye et al, 2012) was used to design primers specific for a given transcript. At least one primer in a pair was designed to span an exon-exon junction to avoid amplification of genomic DNA that might contaminate the samples. All primers (see Table S8) used in this study were tested to only yield a single product and not amplify genomic DNA. The *DmIFT88*, *DmGucy2d*, and several housekeeping genes (e.g., *eIF1A*, *Su(Tpl)*, *TBP*) primers were confirmed to only amplify the respective isoform using cDNA (data not shown). 150 ng of total mRNA were reverse transcribed to cDNA in a reaction volume of 20 μl. The cDNA mix was diluted 1:10, and 4 μl of the dilutions were used in each reaction. iTaq Universal SYBR Green Supermix (Bio-Rad) or THERMO Maxima SYBR Green qRT-PCR Master Mix was used to detect amplification of the PCR products. The reactions were set-up up according to the manufacturer's protocols and run on the CFX384 Touch (Bio-Rad) or Applied Biosystem QuantStudio (Flex 6) Real-Time PCR Detection System (Bio-Rad) (Figs S3C, S4C, and S8B). The total mRNA was isolated three times per genotype, and each sample was measured in triplicates. Results were analysed using RStudio (Rstudio) or Prism (GraphPad).

## Measurements of sound evoked neuronal responses and antennal mechanics

Flies of the respective genotype and age (mentioned on the respective figures and their legends) were immobilised and glued (with a 50/50 mixture bee-wax: paraffin) to a holder. All measurements were recorded from head-fixed flies. The head and non-measured antenna were affixed to the thorax using UV-hardening dental glue, and the reference electrode was positioned in the thorax (for details, see Gopfert et al [2005]).

For maximum compound action potential (μV) (Fig S2D) recording, a tungsten needle was inserted into the antenna's base (i.e., between the head and antenna). The other antenna was immobilised, and the reference electrode was positioned in the thorax.

For maximum mechanical gain and antennal fluctuation powers (kBT) (Figs 2E and 5I), displacements of the free antenna were measured in non-loading condition near the tip of the antennal arista with a scanning laser Doppler vibrometer (Polytec PSV-400, Polytec; Waldbronn) equipped with a DD-500 displacement decoder, with or without external sound stimulation.

To calculate the fluctuation energy of the antennal sound receiver, the mechanical free fluctuations of the receiver were measured in the absence of sound stimulation. Velocity amplitudes were Fourier-transformed to generate a frequency spectrum for the antenna's vibration velocity ($\chi_{(\omega)}$) in the range of 100–1,500 Hz. Velocity amplitudes were subsequently converted into displacement amplitudes ($\chi_{(\omega)} = (\frac{\chi_{(\omega)}}{\omega})$) with $\omega = 2\pi f$ and squared, yielding the power spectral density (PSD) $\chi^2_{(\omega)}$ of the antennal displacement. To describe the fluctuations, the PSD was fitted with the function of a simple harmonic oscillator model, $\chi^2_{(\omega)} = \frac{F_0^2/m^2}{(\omega_0^2 - \omega^2)^2 + (\omega\frac{\omega_0}{Q})}$.

where $F_0$ is the external force, m is the mass (50 ng [Albert et al, 2007]), and $\omega_0$ is the natural angular frequency ($\omega_0 = \sqrt{K_s/m}$, where $K_s$ is the spring constant), and Q is the quality factor ($Q = m\omega_0/\gamma$, where $\gamma$ is the damping factor). When in equilibrium with its surrounding, antenna fluctuates in response to the thermal noise, described as:

$\frac{1}{2}K[X^2] = \frac{1}{2}k_BT$ with $K = K_s$, where K is the effective stiffness, $K_s$ the spring constant, $k_B$ is the Boltzmann constant (1.38 × 10⁻²³ J/K), and T is absolute temperature in Kelvin. To calculate the total energy (power) of the oscillator, we can describe the system as: $E = \frac{K_s}{K}T(k_BT)$, where K is derived from the fit.

To measure the mechanical sensitivity gain flies were stimulated with pure tones at the mechanical eigen-frequency ($f_{eigen}$) for the antenna measured, generated by a loudspeaker close to the animal. Sound particle velocities (SPVs) were recorded by a closely placed pressure-gradient microphone (Emkay NR 3158), as described (Gopfert & Robert, 2002). Flies were stimulated with pure tones covering anan intensity-range of 96 dB, in steps of 6 dB. The displacement amplitudes of the antenna at the stimulus frequency were read out from the displacement spectrum. Mechanical sensitivity was determined by normalising the displacement amplitude to the tone particle velocity. To determine the amplification gain provided by motile responses of Johnston's organ neurons, the sensitivity the antenna reached at low tone particle velocities was normalised to that observed at high particle velocities, and calculated via: *gain = Displacement(m)-ω/SPV(m/s)* with $\omega = 2\pi f_{eigen}$. Mechanical sensitivity is then $gain_{max}/gain_{min}$ for the respective fly. Data were analysed and graphs were generated using RStudio (RStudio) or Prism (GraphPad).

## Sample preparation and image acquisition in transmission electron microscope

The samples for electron microscopy were processed and sectioned as previously described (Jana et al, 2016). Antennae were dissected from the heads and incubated in fixative (2% formaldehyde, 2.5% glutaraldehyde in 0.1 M sodium biphosphate buffer pH 7.2–7.4) overnight. Samples were washed several times in PBS and then incubated for 1.5 h in 1% osmium tetroxide for postfixation. Samples were washed several times in deionised water before incubating them in 2% uranyl acetate for 20 min on a rotator for *en bloc* staining. After this step, samples were dehydrated in a graded sequence of alcohol dilutions (from 50% to a 100%). Ethanol was removed through two incubation steps with propylene oxide for 30 min. Samples were embedded in resin overnight and transferred to a fresh batch of resin for 1 h. Under a dissection microscope, the second segment of the antenna was separated from the third. The second segment was placed in the desired orientation into a mould with resin and polymerised. Samples were sectioned ~70 nm thick, collected on formvar-coated copper slot grids and stained with 2% uranyl acetate followed by lead citrate staining. Finally, samples were air-dried and investigated and photographed at 100 keV or 120 keV using the Hitachi H-7650 or FEI Tecnai T12 transmission electron microscope (Figs S1B, S2E–H, and S4D–F).

### Conditional knockdown in the flies

The Gal4$^{Chat19b}$ driver is active in cholinergic neurons, including chordotonal and olfactory neurons, in all developmental stages of the fly (Salvaterra & Kitamoto, 2001; Jana et al, 2011). Thus, to avoid confounding effects from the ciliogenesis defects and be able to spatiotemporally knockdown our gene of interest in the adults, we adopted a Gal4$^{Chat19b}$(or Gal4$^{Iav}$) TubGal80$^{ts}$-UAS-RNAi-based system. The Gal4 activity was repressed through co-expression of a temperature sensitive version of Gal80 under *Tubulin* promoter (denoted as TubGal80$^{ts}$). At the permissive temperature (18°C) TubGal80ts is functional, acting as a negative regulator by binding to Gal4 and preventing it from attracting polymerase to a UAS element in the fly genome (McGuire et al, 2003). Thus, a candidate (i.e., *DmIFT88, DmGucy2d*) mRNA will only be reduced through RNAi if flies are kept at the restrictive temperature (29°C). Flies were reared at 18°C, shifted to the restrictive temperature after they emerged from pupae (when ciliogenesis is finished) and then submitted to the climbing (negative-gravitaxis) assay for up to 15 d (also see, Fig 2B).

### Climbing (or negative-gravitaxis) assay

To quantify climbing behaviour, a behavioural readout of sensory function, flies were collected in 24-h time windows and kept in groups of the same age. The animals were transferred to vials with fresh food every 3 d. Before the behaviour experiments, flies were separated according to sex into groups of 10 animals that would be tested jointly. Flies were briefly anaesthetised using $CO_2$ for sorting. The flies were kept in the room where the behavioural experiments were conducted for about 1 h before the actual experiment. This time period was given to allow the animals to acclimatise to potentially different light and temperature regimes. Experiments were conducted in measuring cylinders made of glass. Animals were placed in cylinders a couple of minutes before the assay was started. Cylinders were sealed on top with perforated parafilm to prevent the flies from escaping. Cylinders were tapped until all flies were at the bottom of the cylinder, and flies were subsequently allowed to walk up. All experiments were videotaped for 1 min and analysed blindly only once a full data set (including control flies) was obtained. The metric used for analysis was the number of flies above 10 cm after 10 s, as, in the set-up. The flies expressing a hairpin against *mCherry* (UAS-*mCherry*-IR; negative control) in cholinergic neurons need on average 10 s to perform this task. Videos were analysed for the whole duration of 10 s after tapping the cylinder, to rule out that flies reached the 10-cm mark and walked back down again (this protocol is a modified version of the assay used in Jana et al [2021]). Data and statistical analysis, and generation of graphs were done using R Studio (RStudio) or Prism (GraphPAd).

### Cell culture and immunoprecipitation experiments

For co-immunoprecipitation of ectopically expressed proteins (Figs 3A and 4C and E), *D. melanogaster* (Dmel) cultured cells were co-transfected with the corresponding constructs (GFP-tagged at the C-terminus of either *Dm*Gucy2d full-length or fragments [T1–T5];

eGFP-tagged at the C-terminus end of the Iav protein) with 3xHA::*Dm*IFT88. The co-overexpression of GFP alone with 3xHA::*Dm*IFT88 serves as a negative control in these experiments. Cells were seeded at a density of 3 × 10$^6$ cells/well in six-well plates (Orange Scientific). 1 h later, they were transfected using Effectene Transfection Reagent (Quiagen) according to the manufacturer (see Table S8 for primer information). Cells were harvested 3 d after transfection. Cells were resuspended, the suspension was collected and centrifuged and pellets were stored at –80°C, if not processed immediately. Ectopically expressed *Dm*Gucy2d::GFP or Iav::GFP were immunoprecipitated at 4°C for 2 h from Dmel cell lysates (three wells in total for each condition per experiment) using polyclonal anti-GFP antibodies and protein A or G magnetic Dynabeads (Thermo Fisher Scientific). 2 $\mu$g anti-GFP antibodies were added to protein-A or -G magnetic beads and incubated for 30 min at room temperature. Cells were homogenised in lysis buffer: 50 mM Tris–HCl pH 8, 250 mM NaCl, 1 mM DTT, 2% NP-40, 0.5% of SDS, 0.5% sodium deoxycholate, 1× protease inhibitor, 5 yg/ml leupeptin and 15 yg/ml aprotinin, 0.1% digitonin at 4°C for 30 min. Then, benzonase was added to the lysate (final concentration 0.25 U/$\mu$l) and incubated for another 15 min. After centrifugation at 13,500*g* for 20 min at 4°C, the pre-cleared supernatants were incubated with the coupled beads with antibodies. After several washes with lysis buffer, bead pellets were boiled in SDS sample buffer, separated by 4–15% gradient SDS–PAGE and transferred onto LI-COR nitrocellulose membranes for Odyssey. GFP, *Dm*Gucy2d::GFP, Iav::GFP constructs, and HA::*Dm*IFT88 were visualised with anti-GFP or anti-HA antibodies. Secondary Li-COR antibodies IRDye 680RD and IRDye 800 (Li-COR) were used as a second step (see Table S7 for details on the antibodies used).

### Cuticle clearing and imaging of antennae whole-mounts

The cuticle of the adult flies is opaque and autofluorescent, which makes this organ generally unsuitable for imaging. A clearing method that made the cuticle transparent without affecting the morphology of the cells/tissues was developed to image the cytoplasmic GFP expressed in the chordotonal neurons in adult flies with different genotypes. It allowed imaging of the whole antenna without mechanical manipulation. The whole heads were fixed in 4% of PFA in PBS (supplemented with 0.05% Triton-X) for 40 min on ice (Kolesova et al, 2016). Heads were then transferred to a reaction tube containing FocusClear (CelExplorer) and incubated overnight. For imaging, heads were mounted on a slide in MountClear (CelExplorer). Several layers of clear tape were used as spacers between the slide and the coverslip. Images were acquired on a SP5 Upright microscope (Leica). The size of the image stack was chosen to contain the whole antenna segment. Images were then analysed using ImageJ or Imaris v6.4 software (Bitplane).

### Statistical analysis

All statistical analyses were performed using either RStudio (RStudio) or Prism (GraphPad). The details of the software and statistical method used to analyse a given set of experiments are

mentioned in the relevant figure legends. In all figures and source data files, $P > 0.05$ is indicated by "ns" (not significant), and $P \leq 0.05, \leq 0.01, \leq 0.001$, and $\leq 0.0001$ are repseneted by *, **, ***, and **** symbols, respectively.

## Supplementary Information

## Acknowledgements

We thank Daniel F. Eberl, Li E. Cheng, Changsoo Kim, Yun Doo Chung, and Élio Sucena for reagents, and Tejaswini M. and Chaithra D. for help in some experiments. We thank M Bettencourt-Dias and SC Jana Lab members for reviewing the manuscript and providing helpful discussions on the manuscript. We thank the IGC Advanced Imaging unit (and its head, Gabriel G Martins), IGC Electron Microscopy unit (and its Head, Erin Tranfield), NCBS Central Imaging & FACS Facility, and NCBS Electron Microscopy Facility for helping us with image acquisition, and the IGC and NCBS fly facilities for assisting us with fly husbandry. S Werner (SFRH/BD/52176/2013), P Okenve-Ramos (PTDC/BIA-BID/32225/2017), P Priya (201610141058), and SC Jana (SFRH/BPD/87479/2012) are supported by the FCT (Fundação Portuguesa para a Ciência e Tecnologia, Portugal), CSIR (Council of Scientific & Industrial Research, India), and NCBS-Tata Institute for Fundamental Research (TIFR) Fellowships/Grants/Contracts. SC Jana and M Bettencourt-Dias acknowledge FCT (PTDC/BIA-CEL/32631/2017 to SC Jana), TIFR-DAE (Intramural Project Grant to SC Jana), and the European Research Council Consolidator Grant (CoG683528 to M Bettencourt-Dias) for their support through research grants.

### Author Contributions

S Werner: conceptualization, data curation, formal analysis, validation, investigation, visualization, methodology, and writing—original draft, review, and editing.
P Okenve-Ramos: data curation, formal analysis, validation, investigation, visualization, methodology, and writing—original draft, review, and editing.
P Hehlert: data curation, formal analysis, validation, investigation, visualization, and writing—review and editing.
S Zitouni: investigation and visualization.
P Priya: data curation, formal analysis, validation, investigation, visualization, and writing—review and editing.
S Mendonça: investigation and visualization.
A Sporbert: investigation and visualization.
C Spalthoff: investigation and visualization.
MC Göpfert: resources, supervision, funding acquisition, project administration, and writing—review and editing.
SC Jana: conceptualization, resources, data curation, formal analysis, supervision, funding acquisition, validation, investigation, visualization, methodology, project administration, and writing—original draft, review, and editing.
M Bettencourt-Dias: conceptualization, resources, supervision, funding acquisition, project administration, and writing—original draft, review, and editing.

### Conflict of Interest Statement

The authors declare that they have no conflict of interest.

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
