## [Reviewer comments · Life Science Alliance]

Life Science Alliance

IFT88 maintains sensory function by localising signalling proteins along *Drosophila* cilia

Sascha Werner, Pilar Okenve-Ramos, Philip Hehlert, Sihem Zitouni, Pranjali Priya, Susana Mendonça, Anje Sporbert, Christian Spalthoff, Martin Gopfert, Swadhin Jana, and Monica Bettencourt-Dias

DOI: <https://doi.org/10.26508/lsa.202302289>

Corresponding author(s): Monica Bettencourt-Dias, Instituto Gulbenkian de Ciência; Monica Bettencourt-Dias, Instituto Gulbenkian de Ciência; and Swadhin Jana, Instituto Gulbenkian de Ciência

Review Timeline:

Submission Date:	2023-07-25
Editorial Decision:	2023-07-25
Revision Received:	2023-10-24
Editorial Decision:	2023-10-25
Revision Received:	2024-02-07
Accepted:	2024-02-08

Transaction Report:

Please note that the manuscript was previously reviewed at another journal and the reports were taken into account in the decision-making process at *Life Science Alliance*.

Reviews

Reviewer #1 Review

Comments to the Authors (Required):

In this revised version of their manuscript the authors provide several additional experiments that greatly reinforce their conclusions. In particular the EM data combined with physiological recordings for several RNAi conditions strengthen the demonstration that IFT88 is required for maintaining ciliary function but not ciliary architecture. This function is mediated through transport of at least two cargos: IAV and DmGucy2d.

The authors responded positively to all my requests. I no longer have any reservations about publishing this very interesting study.

Minor points:

Comment on Figure 2: The results show only moderate effects on IAV of post-eclosion depletion of IFT88. The amount of IAV is reduced by only 9%, suggesting that IAV turnover is slow in the ciliary compartment or not solely dependent on IFT88. The WB shows several apparently nonspecific bands that are not explained (?) and IAV-GFP appears to be weakly expressed, making these IAV results less convincing than the other parts of this manuscript. However, this does not affect the overall significance of the manuscript.

Sup Fig 7F: there is a strong reduction of fly gravitaxy at 9 days for the cherry control (40% less). I do not understand if there is a difference with experiments of Figure S7E or Figure 2D where control flies were almost not affected?

Sup Fig3E is not cited.

Please update the methods section to fit with the new figure numbering or experiments: example IAV quantifications are now on Figure 3B not 2D. Explain how "Max mechanical gain" and "antennal fluctuation powers" are measured, explain how IFT train length are measured...

Reviewer #2 Review

Comments to the Authors (Required):

In this revised version, Werner et al. strengthen what was already a very nice manuscript. They are now bringing:

- thorough description and quantification of IFT trafficking in Drosophila cilia
- transmission electron microscopy analysis of the cilia in IFT88RNAi cell lines
- identification and characterisation of a new membrane protein (a TRPV channel called Inactive/lav) as a new cargo of the IFT machinery
- direct evidence that this cargo interacts with the IFT88 protein
- electrophysiology data to further characterise the sensory phenotypes

The amount and quality of the new data is really impressive and brings further insights on how IFT88 (and presumably the IFT process) contributes to the maintenance of cilia in Drosophila by contributing to the proper localisation of membrane receptors. It should be pointed out that these cilia represent a powerful model to decipher the role of IFT in long-lived cilia such as in photoreceptors of the retina for example.

Several of these experiments are not trivial and I congratulate the authors for their efforts. I strongly support acceptance of the manuscript that for sure will attract attention in the cilia community at large. Actually, there is so much new information that this might deserve a full article rather than a report, but this decision belongs to the editor (I also know that it takes time to reformat manuscripts).

I have a comment about IFT88. It is remarkable that IFT88 interacts directly with the two cargoes reported here. Since the studies were done in S2 cells where IFT proteins and the cyclase are not expressed, this indicates direct interaction, i.e., in the absence of the IFT complex/train and outside the ciliary compartment. This is quite similar to what was observed in mammalian cells for IFT88 and the leucine-rich repeat containing 56 protein (LRRC56) upon

expression of tagged versions in HEK293 cells (Bonney et al. AJHG 2018). This indicates that the IFT88, presumably via its TPR domains, indeed function to capture a wide variety of cargoes. The authors might want to emphasise this point in the discussion.

Minor comments:

Bottom page 6: the length of IFT trains was first reported by Jordan et al. (NCB2018) using cryo-electron microscopy.

Page 7: no capital for kinesin, please check throughout the manuscript

Top page 9: "hatch" instead of "eclode"?

Page 11: Lee et al. ("l" missing)

S2 cells. What is the nomenclature? There are several versions in the manuscript: "S2-cells" "S-2 cells" "S2 cells", sometimes in italics, sometimes not. Please homogenise.

Top page 13: several organisms ("s" missing)

Bottom page 17: altered instead of alter?

Bottom page 18: that transport of ("s" to be removed)

Legend figure 1: "two examples of merged kymographs"

Reviewer #3 Review

Comments to the Authors (Required):

The authors have added extensive new data in response to the reviewers' comments. Strengths of this work are the demonstration and characterization of IFT88 localization and movement in larval (and adult) mechanosensory neuron cilia in *Drosophila*, the demonstration that IFT88 may be required to maintain cilia sensory functions but not cilia structure following ciliogenesis, and the role of IFT88 in ciliary localization of a TRP channel and a transmembrane guanylyl cyclase.

However, a few issues still remain.

- 1) Some of the data (eg. Figure 2E and related) do not appear to include all the appropriate genetic controls. Canton-S alone is not sufficient
- 2) Unless I misunderstood it, the mRNA quantification data shown are normalized between the mCherry RNAi and the IFT88 RNAi. Shouldn't the appropriate control be before and after the temperature shift? This applies to Supplemental Figures 3E and 4C. These controls are important to demonstrate that the knockdown is specific to the adult stage.
- 3) It also is a bit unclear why the control for comparison is only the mCherry RNAi as opposed to also using the IFT88RNAi strains grown at 18C.
- 4) While it is interesting that IAV and Gucy2d interact with IFT88 in S2 cells, it remains unclear whether these interactions are relevant in vivo
- 5) Despite the authors' best efforts, the evidence that lav and Gucy2d are bona fide cargoes of IFT88 remains quite weak. There are minor effects on localization of each protein in the IFT88 RNAi background and the authors are unable to demonstrate IFT-like movement of either lav or Gucy2d. Moreover, as mentioned above, any direct interaction is demonstrated in a somewhat artificial context. Demonstrating that a protein is an IFT cargo requires extensive experimental proof (only some bona fide cargoes of IFT trains have been identified to date), and the data as presented are not conclusive.
- 6) Finally, the authors are to be commended for the identification of a new function for Gucy2d. However, the parallels drawn with vertebrate photoreceptors are a bit of a stretch. Vertebrate phototransduction clearly uses cGMP as the pivotal second messenger. All current evidence points to mechanotransduction in fly mechanosensory neurons potentially being mediated directly by members of the TRP channel family. cGMP may perhaps play a modulatory role but the role of this second messenger is quite significantly different between these sensory neuron types.

Reviewer #1 Review

Comments to the Authors (Required):

In most organisms cilia assembly and maintenance relies on IFT. The precise role of IFT in various aspects of cilia maintenance is still partially explored. In this study, the authors address for the first time in the *Drosophila* model organism, the function of IFT after cilium assembly. They show by conditional RNAi inactivation using tricky genetic tools that IFT88 is required for proper function of cilia in adult *Drosophila* after cilium assembly. They nicely demonstrate that IFT88 traffics inside cilia using live imaging (only the second time IFT is visualized in *Drosophila* cilia!). They also show that IFT88 interacts with one member of the Guanylyl cyclase family of proteins, DmGucy2d, expressed in chordotonal neurons (ChO), and which accumulation inside the chordotonal cilia requires IFT88. Last, they observe that knock down of DmGucy2d after cilia assembly leads to defects in sensory behaviors of the flies. The data are convincing, support the role of IFT in the maintenance of ciliary function but perhaps only bring partial mechanistic advance on ciliary maintenance. The following points need to be clarified:

Major points:

It is surprising that KD of DmGucy2d gives a stronger climbing phenotype than KD of IFT88: could this be explained by differences in RNAi efficiency (it is not quantified for KD of DmGucy2d)? Could it be because DmGucy2 is also expressed in other cholinergic neurons that could also play a role in locomotor or sensory behaviors? Did the authors repeat the experiment with a more restrictive Gal4 driver (for example *lav-Gal4*)? As well, how leakiness of DmGucy2d RNAi expression at 18°C (Figure 3F) can be explained, as (1) it was not observed for IFT88 using the same Gal4 driver and UAS-reporter constructs/attP insertion platform (if I understand correctly the methods and strain used) and (2) knowing that apparently DmGucy2d is not expressed in all ChO organs? Could the sensory deficits be quantified by electrophysiology in DmGucy2 KD flies, like it was for IFT88 full KD in sup Figure 2C, to strengthen the demonstration that DmGucy2 is indeed required for the maintenance of antennal chordotonal organ function?

The co-immunoprecipitations of Figure 4C could be improved. T4 seems to interact much better than T5, which is surprising and T1 interaction appears also unexpectedly stronger than the interaction with the FL protein. This should at least be discussed. The authors suggest that mutations in the conserved residues of the intracellular domain could lead to defective IFT88 interaction and be responsible for LCA: this is an interesting hypothesis. Did the authors test the interaction with proteins mutated in the few conserved residues for which mutations have been described in LCA (A710V, I734A, R768W) and that fall in the domain that is present in T4 but not in T5, thus potentially involved in the proposed interaction? This could better support their hypothesis regarding the etiology of LCA. It should also be noted that these IPs do not demonstrate a direct interaction between IFT88 and DmGucy2.

Minor points :

-An interesting ciliary phenotype is quantified: the bending of the cilia at its base. The origin of such defect is not clear but could it be related to motility defects of the chordotonal cilia (as suggested in the discussion)? Is it technically conceivable to measure this as described by one of the co-authors (Göpfert, M. C. & Robert, D. Proc. Natl. Acad. Sci. USA 100, 5514-5519 (2003). This could be very interesting even if it may fall behind the scope of this manuscript.

-Are there other expression domains for CG34357 than only the PNS ?

-The authors postulate that late onset ciliopathies cannot be explained by defects in cilia assembly (Introduction page 3, sentence « while defects in cilia assembly can lead...»). I think this is over-simplification, as tissue maintenance (which could explain late disease onset or progressive degeneration) is confused here with cilium maintenance. Progressive tissue degeneration can be due to defects in cilium assembly. For instance, impaired cilia assembly in cells only involved in tissue repair could explain late disease onset.

-The authors conclude that defects in DmGucy2 and IFT88 interactions could explain several ciliopathies in humans. This is an hypothesis and the results do not demonstrate this formally, so this conclusion should be modulated at

least in the abstract.

-Page 5 last sentence : this study and that of Lee et al (2018), both look at ChO organs, so I suggest to remove « cell type » in the sentence.

-Page 10 last sentence « in » the cilium instead of « to »?

-Page 11 :

Sentence : « suggesting that defective transport of Gucy2d into photoreceptor cilia... » . It's a bit of an overinterpretation?

Sentence « our finding identifies a novel mechanism for cilia maintenance » : is it really novel ?

-Page 13 : « these observations suggest that some of the phenotypes of LCA patients arise... » again seems a bit of an overinterpretation to me

-Figure 1C : explain what the arrowheads point to.

-Figure 2D : recall in the Figure legend at which day are made the quantifications.

-It is not clear which GFP antibody is used for the IP.

Reviewer #2 Review

Comments to the Authors (Required):

In this manuscript, Werner and co-workers ask the import question of the role of intraflagellar transport in the maintenance of long-lived cilia, using *Drosophila* as model organism.

They first demonstrate movement of the landmark IFT88 protein in cilia of chordotonal neurons in larval stages and then use IFA to confirm that IFT proteins are still expressed and located to cilia in adult stages, so well after their construction.

Next, they used a temperature-inducible RNAi knockdown with appropriate controls to deplete IFT88 and monitored the impact on cilia morphology and function. Although the structure does not seem affected, defects in fly climbing behaviour were observed, revealing that IFT is required for proper function.

In an effort to decipher the molecular mechanisms behind this phenotype, they screened for several potential effectors and identified a family of guanylyl cyclases as promising candidates. RNAi knockdown revealed that these were involved in climbing behaviour and further investigation showed that at least one of them needs the presence of IFT88 to be properly localised. Strikingly, this cyclase interacts with IFT88 in an in-cellulo assay, leading to an elegant model where IFT88 would contribute to the transport of the enzyme to the cilium.

Overall, this is a very nice study, cleverly designed and carefully executed. Although the figures are complex with multiple panels, they are well organised and follow an impeccable logic that allows getting the main message through. This study will be of interest for cell biologists but also for clinicians studying long-lived cilia such as in photoreceptors. The message could be quite significant in this context, as highlighted by the authors in the last chapter. I only have one question about the interpretation of the data, but this does not condition the acceptance of the manuscript.

Question: the main title says that IFT88 'transports' Gucy2d. I would be interested to know what type of transport are the authors referring to? Is it (1) IFT itself (i.e. movement along cilia as shown for tubulin in *Chlamydomonas* for example) (2) transport from the cytoplasm to the base of cilia or (3) entry in cilia?

In principle, the interaction between IFT88 and Gucy2D could support any of these proposals. In the case of IFT though, one has to keep in mind that IFT88 is not alone but in a complex, so might not be easily available for interaction (Taschner et al, JCB2014). The situation might be different in *Drosophila* however, since almost one half

of the IFT-B1 complex seems to be missing compared to other organisms, including the IFT74/81 supposed to be essential for tubulin transport.

Minor comments.

I would add a few words in the abstract to say that IFT is not required for maintenance of the ciliary structure.

Page 3. "nephronophthisis" instead of "Nephronophthisis"; "Alström" instead of "Alstrom"

Bottom page 3. Please note that several matrix proteins (PKA regulatory protein, kinesin 9B) were mis-localised following depletion of IFT88 in Trypanosoma

Page 4. DmlIFT88-containing trains

Top page 6. Anterograde trains do indeed contain more IFT proteins (Buisson et al JCS2013; Chien et al Elife 2017) than retrograde trains but the compaction is different. Correlative light and electron microscopy revealed that they actually have a similar length around 200 nm (Stepanek & Pigino Science 2016), at least in Chlamydomonas. This is in contrast to the original publication (Pigino et al JCB2009), presumably because of the presence of arrested trains.

Page 7. "The Gal4Chat19b driver expresses in the peripheral and central nervous system in all developmental stages." It seems that one word is missing here

Top page 8. "rise" instead of "raise"

Second half page 10. For T1 localisation, figure 5A should be called instead of 4C

Reviewer #3 Review

Comments to the Authors (Required):

This paper addresses the interesting issue of whether IFT is required for ciliary maintenance in the model system type I sensory ciliated neurons in Drosophila. The authors find that the conserved protein IFT88 moves along sensory cilia in Drosophila larvae, and immuno-staining indicates that this movement continues in the adult fly, suggesting that it may be needed for cilium maintenance. Acute knockdown of IFT88 in the adult affects sensory behaviors, including climbing, and has subtle effects on cilia morphology. They identify the guanylyl cyclase Gucy2d as an IFT88 cargo that is important for the sensory functions of chordotonal neurons, and further demonstrate binding of IFT88 to the intracellular portion of Gucy2d.

Overall, the experiments are well done and the findings are interesting. However, there are essentially two stories in this paper, neither of which is fully developed. The first has to do with demonstrating a role for IFT88 in cilia maintenance, and the second describes identification of the GC as a new molecule involved in climbing behavior that interacts with IFT88. Some suggestions on how this work might be expanded to flesh out these stories a bit better are provided below.

Major Comments

- 1) Retrograde movement in Figure 1Biii -This is difficult to see. The diagram to the left helps, but it does not match up with the actual images to the right in terms of its width. Please redraw the diagram to scale so that it's clear which IFT trains they correspond to (both anterograde and retrograde movement).
- 2) Can the authors comment on what might account for the heterogeneity in retrograde velocity?
- 3) What does it mean if the base of the cilium is curved? This finding is not explored further. Does the curved morphology directly influence the sensory functions of the cilium/neuron?
- 4) For quantification of IFT88 expression levels in Supplemental Figure 2B, it is also important to verify +/- temperature shift that the IFT88 hairpin provides a specific knockdown of adult IFT88 levels using the conditional promoter system. Moreover, please examine levels of at least one other IFT protein to establish specificity.

4) Does prolonged temperature shift with IFT88 temperature-sensitive knockdown resemble the KO mutant so that cilia don't form? Quantification of mRNA levels with the conditional knockdown and assessment of long-term temperature shift would strengthen the idea that the subtle effects on morphology and behavior are due to requirements of IFT88 for ciliary maintenance (and not, for example, due to partial *lof*).

5) Please describe the statistical tests used clearly in the figure legends.

6) There is no attempt to connect the newly identified GC with the mechanosensory channels. How might a GC function to mediate mechanosensation? Does loss of the GC result in altered localization or loss of IAV for instance?

6) It seems important for the story to demonstrate that DmGucy2d moves via IFT in larvae and in the adult

7) For the KD of Gucy2d in adults: Does knockdown affect cilia morphology? Bending phenotype?

8) Authors conclude KD of IFT88 is important for Gucy2d localization to affect behavior. But this effect on Gucy2d intensity (Figure 5C) is really minor. In fact, the effects are really not that much stronger than the effects on localization of IAV. Does IFT88 also bind IAV to help localize it? Do you hypothesize that additional guanylyl cyclases (2 others looked at had climbing defects) are important for this behavior as well? Would be interesting to examine whether KD of IFT88 affects their ciliary localization. In other words, while the authors demonstrate that IFT88 interacts with this GC, it is not at all clear that this is the most important and relevant role for this protein.

Minor Comments

1) I don't believe that signal intensity on its own is the best readout for train length

2) The authors mention that entry of SSTR3 into cilia is IFT-independent. In fact, it has been demonstrated that SSTR3 entry into cilia is TULP3 and IFT-A-dependent.

July 25, 2023

Re: Life Science Alliance manuscript #LSA-2023-02289-T

Dr. Swadhin Chandra Jana
Instituto Gulbenkian de Ciência
Rua da Quinta Grande 6
Oeiras 2780156
Portugal

Dear Dr. Jana,

Thank you for submitting your manuscript entitled "IFT88 maintains sensory function by localising several signalling proteins along *Drosophila* cilia" to Life Science Alliance. We invite you to submit a revised manuscript addressing the following Reviewer comments:

- Address Reviewer 1's minor points.
- Address Reviewer 2's minor comments.
- Address Reviewer 3's points #1-3 & 6.

Thank you for this interesting contribution to Life Science Alliance. We are looking forward to receiving your revised manuscript.

Sincerely,

- A letter addressing the reviewers' comments point by point.
- An editable version of the final text (.DOC or .DOCX) is needed for copyediting (no PDFs).
- High-resolution figure, supplementary figure and video files uploaded as individual files: See our detailed guidelines for preparing your production-ready images, <https://www.life-science-alliance.org/authors>
- Summary blurb (enter in submission system): A short text summarizing in a single sentence the study (max. 200 characters including spaces). This text is used in conjunction with the titles of papers, hence should be informative and complementary to the title and running title. It should describe the context and significance of the findings for a general readership; it should be written in the present tense and refer to the work in the third person. Author names should not be mentioned.
- By submitting a revision, you attest that you are aware of our payment policies found here: <https://www.life-science-alliance.org/copyright-license-fee>

B. MANUSCRIPT ORGANIZATION AND FORMATTING:

POINT-BY-POINT answers to the reviewers' comments:**Reviewer #1:**

In this revised version of their manuscript the authors provide several additional experiments that greatly reinforce their conclusions. In particular the EM data combined with physiological recordings for several RNAi conditions strengthen the demonstration that IFT88 is required for maintaining ciliary function but not ciliary architecture. This function is mediated through transport of at least two cargos: IAV and DmGucy2d.

The authors responded positively to all my requests. I no longer have any reservations about publishing this very interesting study.

Authors-01: We are very happy that the reviewer appreciated our sincere efforts and recommended our work for publication.

MINOR POINTS:

**- Comment on Figure 2: The results show only moderate effects on IAV of post-eclosion depletion of IFT88. The amount of IAV is reduced by only 9%, suggesting that IAV turnover is slow in the ciliary compartment or not solely dependent on IFT88. The WB shows several apparently nonspecific bands that are not explained (?) and IAV-GFP appears to be weakly expressed, making these IAV results less convincing than the other parts of this manuscript. However, this does not affect the overall significance of the manuscript. (Reviewer here actually refers to Figure 3)*

Authors-02: We agree with the reviewer's suggested hypotheses for the resulting low amount of IAV reduction upon *DmIFT88* knockdown in the adult cilia. Our results point to various possibilities; for instance, IAV turnover in the axoneme is low/slow or a *DmIFT88*-independent mechanism could also be involved in maintaining some fraction of axonemal IAV. We have now discussed these possibilities in the "Discussion" section of the revised manuscript (See Page 19 Lines 11-17).

Regarding the WBs with bands related to IAV::GFP, all three independent co-IP repeats done led to the same 3 α -GFP bands in the co-IP experiment, from which the middle one should be the corresponding IAV::GFP (based on predicted MW and the band observed in the input lane). This pUAS-IAV::GFP vector was kindly provided by Dr. Yun Doo Chung, one of the authors of the paper that previously used it to generate the UAS-IAV::GFP flies showing the expected localisation in the antenna (Gong et al. 2004), corroborated by the antibody also used in this study. Post-transcriptional modifications have been previously found in TRP channels, including phosphorylation or N-linked glycosylation in TRPV channels, the orthologues of IAV (Voolstra and Huber, 2014). These can modify their location, properties and gating, and could explain, if occurred in *Drosophila* IAV protein, the different band sizes. In

fact, phosphorylation of TRP and TRPL channels in multiple sites has been reported in *Drosophila*, modulating their function in phototransduction (Popescu et al., 2006; Voolstra and Huber, 2014). Since the modifications mentioned above would likely increase the protein size, we believe that the observed additional lower bands can be due to unwanted but possible protein degradation during the experimental process. In relation to the weakness of the GFP band, it is important to note that these cells are not stable cell lines, but transiently transfected. This results in a relatively low number of cells expressing *lav::GFP*, and even a smaller percentage of cells expresses both *lav* and *DmIFT88*. For that reason, the amount of protein is not high but enough to find the binding of the two proteins by co-IP. We have now clarified these aspects in the main text (Page 11 Lane 23) and the legend of Figure 3A.

*- *Sup Fig 7F: there is a strong reduction of fly gravitaxis at 9 days for the cherry control (40% less). I do not understand if there is a difference with experiments of Figure S7E or Figure 2D where control flies were almost not affected?*

Authors-03: Indeed, as the reviewer pointed out, there were differences in the results obtained and plotted in Figure 2D vs old Supplemental Figure 7E in control flies at 29°C.

The case of the old Supplemental Figure 7E graph was an additional experiment performed to confirm that the downregulation of *DmGucy2d* in the adult induces negative-gravitaxis defects, using another *DmGucy2d* RNAi line generated independently of RNAi line (BSC, TRiP #28524) used in Figure 5H. For this experiment, we used a different *DmGucy2d* RNAi line (VDRC-KK, #105185) that were obtained from VDRC Stock Center. In this case, the control flies and experimental set up were different to the ones of the rest of the manuscript. In addition, the *mCherry* RNAi line was kept for a long time in our stocks and might have accumulated unwanted and adverse genetic variations that changed behavioural results. Despite that, the results indicated, as expected, that decreasing *DmGucy2d* RNA amounts impairs sensory function in adulthood.

However, following the reviewer's (and our) concern with the results, we repeated the experiments in a similar set-up as the original experiments (described in the rest of the manuscript) using a newly acquired *mCherry* RNAi fly from Bloomington Stock Center (BSC, # 35785). The graph shows a clearer and more significant impairment on negative-gravitaxis upon *DmGucy2d* RNA reduction only during adulthood using a VDRC-KK *DmGucy2d* RNAi line ((VDRC-KK, #105185) referred to as *DmGucy2d-IR2*) (see new Supplemental Figure 7D).

*- *Sup Fig3E is not cited.*

Authors-04: We apologise for this unintended error. We cited this information in the new manuscript accordingly in Page 9 Line 27.

**- Please update the methods section to fit with the new figure numbering or experiments: example IAV quantifications are now on Figure 3B not 2D. Explain how "Max mechanical gain" and "antennal fluctuation powers" are measured, explain how IFT train length are measured...*

Authors-05: We thank the reviewer for letting us know about the non-updated numbering in the "methods" section as well as description of a few measurements (as pointed by the reviewer) in the "Materials and Methods". Some examples of the updated description are: how "IFT train length", "Max mechanical gain" and "antennal fluctuation powers" are measured are described in Page 26 Lines 12-17, Page 27 Line 17, and Page 28 Line 7, respectively.

Reviewer #2:

In this revised version, Werner et al. strengthen what was already a very nice manuscript. They are now bringing:

- thorough description and quantification of IFT trafficking in Drosophila cilia*
- transmission electron microscopy analysis of the cilia in IFT88RNAi cell lines*
- identification and characterisation of a new membrane protein (a TRPV channel called Inactive/lav) as a new cargo of the IFT machinery*
- direct evidence that this cargo interacts with the IFT88 protein*
- electrophysiology data to further characterise the sensory phenotypes*

The amount and quality of the new data is really impressive and brings further insights on how IFT88 (and presumably the IFT process) contributes to the maintenance of cilia in Drosophila by contributing to the proper localisation of membrane receptors. It should be pointed out that these cilia represent a powerful model to decipher the role of IFT in long-lived cilia such as in photoreceptors of the retina for example.

Several of these experiments are not trivial and I congratulate the authors for their efforts. I strongly support acceptance of the manuscript that for sure will attract attention in the cilia community at large. Actually, there is so much new information that this might deserve a full article rather than a report, but this decision belongs to the editor (I also know that it takes time to reformat manuscripts).

Authors-06: We are glad the reviewer noticed the improvement of the manuscript and the new data added to this new version of the manuscript.

I have a comment about IFT88. It is remarkable that IFT88 interacts directly with the two cargoes reported here. Since the studies were done in S2 cells where IFT proteins and the cyclase are not expressed, this indicates direct interaction, i.e., in the absence of the IFT complex/train and outside the ciliary compartment. This is

quite similar to what was observed in mammalian cells for IFT88 and the leucine-rich repeat containing 56 protein (LRRC56) upon expression of tagged versions in HEK293 cells (Bonney et al. AJHG 2018). This indicates that the IFT88, presumably via its TPR domains, indeed function to capture a wide variety of cargoes. The authors might want to emphasise this point in the discussion.

Authors-07: We highly appreciate the reviewer's input. We have now included these remarks into the discussion (See Page 19 Lines 28-30).

MINOR COMMENTS:

**- Bottom page 6: the length of IFT trains was first reported by Jordan et al. (NCB2018) using cryo-electron microscopy.*

Authors-08: We thank the reviewer for giving us this information. We added the appropriate reference to the manuscript (See Page 6 Line 32).

**- Page 7: no capital for kinesin, please check throughout the manuscript*

Authors-09: We modified the capital kinesin throughout the manuscript as requested by the reviewer (See Page 4 Line 1, Page 7 Lines 18 and 20, and Page 8 Line 27).

**- Top page 9: "hatch" instead of "eclode"?*

Authors-10: Thanks for raising this point. Since, in the fruit fly researchers' community, the word "hatching" is used for the process of maggots/larvae coming out of eggs, and here we followed the freshly emerged flies, we used "flies emerge from pupae" (See Page 9 Line 6, Page 16 Line 26, Page 26 Line 21, and Page 29 Line 21).

**- Page 11: Lee et al. ("I" missing)*

Authors-11: The noticed typo was modified as requested (See Page 11 Line 13).

**- S2 cells. What is the nomenclature? There are several versions in the manuscript: "S2-cells" "S-2 cells" "S2 cells", sometimes in italics, sometimes not. Please homogenise.*

Authors-12: We agree with the reviewer that the nomenclature is not consistent throughout the manuscript, and thus we now refer to these cells as "Dmel" cells in the new manuscript (See Page 11 Line 17, Page 11 Line 20, Page 14 Line 8, Page 14 Line 10, Page 14 Line 34, Page 19 Line 31, Page 30 Line 26, Page 31 Line 43, Page 47 Line 6, Page 49 Line 16 and Page 50 Line 6).

*- Top page 13: several organisms ("s" missing)

Bottom page 17: altered instead of alter?

Bottom page 18: that transport of ("s" to be removed)

Legend figure 1: "two examples of merged kymographs"

Authors-13: We thank the reviewer for letting us know about the above typos. These have been corrected on the new manuscript.

Reviewer #3:

The authors have added extensive new data in response to the reviewers' comments. Strengths of this work are the demonstration and characterization of IFT88 localization and movement in larval (and adult) mechanosensory neuron cilia in Drosophila, the demonstration that IFT88 may be required to maintain cilia sensory functions but not cilia structure following ciliogenesis, and the role of IFT88 in ciliary localization of a TRP channel and a transmembrane guanylyl cyclase.

Authors-14: We thank the reviewer for appreciating our efforts and the new data included in the previous version.

However, a few issues still remain.

*- **1)** *Some of the data (eg. Figure 2E and related) do not appear to include all the appropriate genetic controls. Canton-S alone is not sufficient.*

Authors-15: We agree with the reviewer that another genetic control would be a better approach for these electrophysiology experiments. For that reason, we now included, following the rest of experiments' controls, the electrophysiological response measurements from *mCherry* RNAi (crossed to Gal4^{Chat19b}TubGal80^{ts}) flies. Importantly, even new measurements/results do not modify our previous conclusions: acute knockdowns of *DmIFT88* (Figure 2D, E (new)), and *DmGucy2d* (Figure 5H, I (new)) in the sensory neurons of adult flies impair sensory cilia maintenance function.

*- **2)** *Unless I misunderstood it, the mRNA quantification data shown are normalized between the mCherry RNAi and the IFT88 RNAi. Shouldn't the appropriate control be before and after the temperature shift? This applies to Supplemental Figures 3E and 4C. These controls are important to demonstrate that the knockdown is specific to the adult stage.*

Authors-16: As the reviewer correctly points out, the mRNA quantifications are normalised to *mCherry* RNAi. Due to our focus on cilia maintenance research, we use the TubGal80^{ts} system and specific drivers for sensory neurons (Gal4^{Chat19b}) in the antennae of the adult in our experiments. In order to make sure we separate cilia maintenance from ciliogenesis, we only shift the flies from 18°C (from Gal4 inactive)

to 29°C (to Gal4 active), after the animals emerge from the pupae. At that point, cilia of our interest are fully formed (ciliogenesis in these cells occurs during pupal stages -see new Figure 5A and Supplemental Figure 3A, 4A, 7A and 8A for clarification of stages and drivers used in our various experiments related to this concern). For the sensory behaviour analysis experiments (e.g., negative-geotaxis assay), we indeed checked whether the flies without *DmIFT88*- or *DmGucy2d*- RNA knockdown (but, carried respective UAS-RNAi) as they are kept at 18°C throughout their life, and found that those flies show near normal behaviour (see Supplemental Figure 3B and 7B, and **Authors-17** answer).

However, unfortunately, for an RNA level measurement experiment, this control (flies grown at 18°C) suggested by the reviewer would not be the most appropriate one, as change in temperature at which flies are grown/kept would change all biological processing, including RNA expression and stability, leading to difficulty in normalising the levels of quantified RNAs. Therefore, we chose controls that have similar genetic backgrounds and have gone through the same environmental conditions and changes (See Figure 2B, 5A and Supplemental Figure 3A, 4A, 7A and 8A for clarification of stages and drivers used in our various experiments related to this concern).

**- 3) It also is a bit unclear why the control for comparison is only the mCherry RNAi as opposed to also using the IFT88RNAi strains grown at 18C.*

Authors-17: We agree with the reviewer that 18°C experiments are useful, especially to check for the leakiness of the drivers and the specificity of the behavioural phenotypes. With that in mind, we provided graphs of the experiments performed at 18°C in the Supplemental Figures 3B and 7B (*DmIFT88* RNAi and *DmGucy2d* RNAi, respectively) (also see **Authors-16** answer).

4) While it is interesting that IAV and Gucy2d interact with IFT88 in S2 cells, it remains unclear whether these interactions are relevant in vivo

Authors-18: The reviewer is right in the fact that the interaction experiments were performed in a near *in vitro* set up. The fact that Dmel cells do not normally express all three (*Iav*, *DmGucy2d* and *DmIFT88*) genes, actually reinforces our conclusions of the co-IPs, as pointed out and agreed by reviewer #2 (see **Authors-08** answer, and the remarks (in Page 19 Line 28 on (Bonney et al., 2018)) in the new manuscript, as suggested by the reviewer #2).

However, we agree that this biochemical interaction data alone is not sufficient to claim the *in vivo* relevance of the interactions. This comes from the significant decrease of *Iav* (Figure 3B) and *DmGucy2d* (Figure 5D-G) protein amounts along the cilia upon *DmIFT88* RNAi, and the fact that the conditional knockdown of all candidates in the adult display similar sensory function defects (Figure 2D, E and 5H,

l). We highlight that in the case of *DmGucy2d*, there is up to a ~38% reduction of the protein amount in the proximal cilium (Figure 5F) upon a ~30% decrease of *DmIFT88* RNA levels (Supplemental Figure 3C) (see Page 15 Lines 28-33, Page 19 Lines 12-17, and Page 19 Lines 30-32).

5) Despite the authors' best efforts, the evidence that lav and Gucy2d are bona fide cargoes of IFT88 remains quite weak. There are minor effects on localization of each protein in the IFT88 RNAi background and the authors are unable to demonstrate IFT-like movement of either lav or Gucy2d. Moreover, as mentioned above, any direct interaction is demonstrated in a somewhat artificial context. Demonstrating that a protein is an IFT cargo requires extensive experimental proof (only some bona fide cargoes of IFT trains have been identified to date), and the data as presented are not conclusive.

Authors-19: Unfortunately, as the reviewer points out, we were unable to image *in vivo* *DmGucy2d::GFP* particle movement (Page 15 Lines 4-25), nor *lav* in larvae (data not shown). This restricted us to claim that when they move along cilia, they do it at a similar speed of *DmIFT88*. We mentioned this limitation in the “Results” section (Page 15 Lines 23-25) and Figure legend of Supplemental Figure 6. Of note, besides other explanations related to experimental difficulties (such as low *DmGucy2dGFP* intensity), both cargo proteins might have a very slow turnover, which would result in very low frequency of movement (and thus only a particle will move in irregular intervals in a long space of time) which might not be enough for us to catch in the 2-3 minutes of the recordings.

Although the binding assays were done *in vitro*, the fact that *Dmel* cells do not normally express these genes makes us more confident in the specificity of our co-IP experiments (See Page 19 Lines 30-32). *In vivo*, *lav*, *DmGucy2d* and *DmIFT88* localize along the adult fly cilia (Figure 2A, 3B, and 4B). The fact that several TRPs were shown to move within cilia in a fashion similar to IFT in other species, *Gucy2d* was implicated as an IFT88 cargo in mice cultured cells, and both signalling proteins *DmGucy2d* and *lav* are evolutionarily conserved, strengthens our claim of both proteins being *DmIFT88* cargoes in *Drosophila*. Nonetheless, we modified the title of the paper to accommodate these concerns, as well as other sections along the main text (e.g., Page 15 Lines 1-2, and Page 19 Lines 30-32).

**- 6) Finally, the authors are to be commended for the identification of a new function for Gucy2d. However, the parallels drawn with vertebrate photoreceptors are a bit of a stretch. Vertebrate phototransduction clearly uses cGMP as the pivotal second messenger. All current evidence points to mechanotransduction in fly mechanosensory neurons potentially being mediated directly by members of the TRP*

channel family. cGMP may perhaps play a modulatory role but the role of this second messenger is quite significantly different between these sensory neuron types.

Authors-20: Until now, little is known about mechanotransduction in the fly cilia, and as the reviewer indicated, TRPs (and Ca^{2+}) are the best-known components to be most probably involved in this process. Therefore, it is only possible to comment on this subject with appropriate experiments. We could only speculate that *DmGucy2d* might be involved in regulating the levels of the secondary messengers in the cilia compartment, including either cGMP directly (which can further be directly used by ion channels, PKGs or PDEs as described) and/or some other messengers in an indirect way.

Indeed, cGMP is an essential second messenger important for several physiological processes in the lung, blood, heart, intestine, and neural tissues. Second segment antenna *Drosophila* Chordotonal Organs are important for hearing sensation. In this regard, apart from the described retina diseases upon *Gucy2d* mutations, cGMP signalling also has multiple functions described in the auditory processing. For instance, specific transmembrane GCs play a role in the basal integrity of outer hair cells in mice, and GCs are suggested to be protective of hearing upon acoustic trauma or ageing (Marchetta et al., 2020). It is also possible that the produced cGMP regulates as described the synthesis of the signalling molecule cADPR (Graeff et al., 1998), and this indirectly modifies intracellular Ca^{2+} by several means, such as gating the TRPM2 cation channel (Yu et al., 2019).

Our manuscript does not attempt to explain the molecular role of cGMP or GCs in mechanotransduction in the adult fly's chordotonal organ; this would be an entirely new project. However, in this context, our manuscript does show in the first place that there is a certain degree of conservation of the *DmGucy2d* protein between several animal species (Supplemental Figure 9A and B). On the other hand, this manuscript provides a new hypothesis to explain the phenotypes of patients with described mutations of LCA retina disease. We found that several of the human mutations are conserved in the *Drosophila* GC protein (Supplemental Figure 9C), and many are specifically located in the *Dm*IFT88-binding part of the protein, suggesting there could be defective transport of the particulate cyclase in the fully formed cilia membrane of human patients.

REFERENCES (for response to reviewers)

- Bonnefoy, S., C.M. Watson, K.D. Kernohan, M. Lemos, S. Hutchinson, J.A. Poulter, L.A. Crinnion, I. Berry, J. Simmonds, P. Vasudevan, C. O'Callaghan, R.A. Hirst, A. Rutman, L. Huang, T. Hartley, D. Gynspan, E. Moya, C. Li, I.M. Carr, D.T. Bonthron, M. Leroux, C. Care4Rare Canada, K.M. Boycott, P. Bastin, and E.G. Sheridan. 2018. Biallelic Mutations in LRRC56, Encoding a Protein Associated with Intraflagellar Transport, Cause Mucociliary Clearance and Laterality Defects. *American journal of human genetics*. 103:727-739.
- Graeff, R.M., L. Franco, A. De Flora, and H.C. Lee. 1998. Cyclic GMP-dependent and -independent effects on the synthesis of the calcium messengers cyclic ADP-ribose and nicotinic acid adenine dinucleotide phosphate. *J Biol Chem*. 273:118-125.
- Marchetta, P., D. Mohrle, P. Eckert, K. Reimann, S. Wolter, A. Tolone, I. Lang, M. Wolters, R. Feil, J. Engel, F. Paquet-Durand, M. Kuhn, M. Knipper, and L. Ruttiger. 2020. Guanylyl Cyclase A/cGMP Signaling Slows Hidden, Age- and Acoustic Trauma-Induced Hearing Loss. *Front Aging Neurosci*. 12:83.
- Popescu, D.C., A.J. Ham, and B.H. Shieh. 2006. Scaffolding protein INAD regulates deactivation of vision by promoting phosphorylation of transient receptor potential by eye protein kinase C in *Drosophila*. *J Neurosci*. 26:8570-8577.
- Voolstra, O., and A. Huber. 2014. Post-Translational Modifications of TRP Channels. *Cells*. 3:258-287.
- Yu, P., Z. Liu, X. Yu, P. Ye, H. Liu, X. Xue, L. Yang, Z. Li, Y. Wu, C. Fang, Y.J. Zhao, F. Yang, J.H. Luo, L.H. Jiang, L. Zhang, L. Zhang, and W. Yang. 2019. Direct Gating of the TRPM2 Channel by cADPR via Specific Interactions with the ADPR Binding Pocket. *Cell Rep*. 27:3684-3695 e3684.

October 25, 2023

RE: Life Science Alliance Manuscript #LSA-2023-02289-TR

Dr. Swadhin Chandra Jana
Instituto Gulbenkian de Ciência
Rua da Quinta Grande 6
Oeiras 2780156
Portugal

Dear Dr. Jana,

Thank you for submitting your revised manuscript entitled "IFT88 maintains sensory function by localising signalling proteins along *Drosophila cilia*". We would be happy to publish your paper in Life Science Alliance pending final revisions necessary to meet our formatting guidelines.

- please upload all figure files as individual ones, including the supplementary figure files; all figure legends should only appear in the main manuscript file
- please add ORCID ID for the secondary corresponding author -- they should have received instructions on how to do so
- please add the Twitter handle of your host institute/organization as well as your own or/and one of the authors in our system
- please make sure the author order in your manuscript and our system match
- please use the [10 author names et al.] format in your references (i.e., limit the author names to the first 10)
- please add your main, supplementary figure, and table legends to the main manuscript text after the references section
- we encourage you to revise the figure legend for Figure S6 such that the figure panels are introduced in alphabetical order and match the panels in the figure
- please label the Supplementary tables as Tables S1, S2, etc...
- remove tables from the manuscript text and upload them separately in editable .doc or excel format

A. FINAL FILES:

B. MANUSCRIPT ORGANIZATION AND FORMATTING:

Sincerely,

February 8, 2024

RE: Life Science Alliance Manuscript #LSA-2023-02289-TRR

Dr. Monica Bettencourt-Dias
Instituto Gulbenkian de Ciência
Cell Cycle Regulation Lab
Rua da Quinta Grande, 6
Oeiras 2780-156
Portugal

Dear Dr. Bettencourt-Dias,

Thank you for submitting your Research Article entitled "IFT88 maintains sensory function by localising signalling proteins along *Drosophila* cilia". It is a pleasure to let you know that your manuscript is now accepted for publication in Life Science Alliance. Congratulations on this interesting work.

DISTRIBUTION OF MATERIALS:

Again, congratulations on a very nice paper. I hope you found the review process to be constructive and are pleased with how the manuscript was handled editorially. We look forward to future exciting submissions from your lab.

Sincerely,
